# FRAGSEL: FRAGMENTED SELECTION FOR NOISY LABEL REGRESSION

## ABSTRACT

As with many other problems, real-world regression is plagued by the presence of noisy labels, an inevitable issue that demands our attention. Fortunately, much real-world data often exhibits an intrinsic property of continuously ordered correlations between labels and features; where data points with similar labels are also represented with closely related features. In response, we propose a novel approach named FragSel wherein we collectively model the regression data by transforming them into disjoint yet contrasting fragmentation pairs. This allows us to train more distinctive representations, enhancing our ability to tackle the issue of noisy labels. Our FragSel framework subsequently leverages a mixture of neighboring fragments to discern noisy labels through neighbor agreement within both the prediction and representation spaces. To underscore the effectiveness of our framework, we extensively perform experiments on four benchmark datasets of diverse domains, including age prediction, price prediction, and music production year estimation. Our approach consistently outperforms thirteen state-of-the-art baselines, being robust against symmetric and random Gaussian label noise.

## 1 INTRODUCTION

Regression is an important task in many disciplines such as finance (Zhang et al., 2017b; Wu et al., 2020c), medicine (de Vente et al., 2021; Tanaka et al., 2022), economics (Zhang et al., 2022), physics (Sia et al., 2020; Doi et al., 2022), geography (Liu et al., 2023) and more. However, real-world regression labels are prone to being corrupted with noise, making it an inevitable problem we must overcome in practical applications. In previous research, noisy label regression has been primarily studied in age estimation with noise incurred from Web data crawling (Rothe et al., 2018; Yiming et al., 2021). Beyond that, the issues of continuous label errors have also been reported in the tasks of object detection (Su et al., 2012; Ma et al., 2022) and pose estimation (Geng & Xia, 2014) as well as measurements in hardware systems (Zhou et al., 2012; Zang et al., 2019).

The vast amount of noisy label learning research has focused more on classification than regression. Some notable approaches include regularization (Wang et al., 2019; Zhang & Sabuncu, 2018), data re-weighting (Ren et al., 2018; Shen & Sanghavi, 2019), training procedures (Jiang et al., 2018), transition matrix (Yao et al., 2020; Xia et al., 2020), contrastive learning (Zhang et al., 2021a; Li et al., 2022b), refurbishing (Song et al., 2019) and sample selection (Lee et al., 2018; Ostyakov et al., 2018). Particularly, sample selection can be further divided into exploring the memorability of neural networks (Arpit et al., 2017; Zhang et al., 2017a) and delineating samples via the loss magnitude (Wei et al., 2020). To the best of our knowledge, two works address the noisy label problem for regression. Garg & Manwani (2020) propose an ordinal regression-based loss correction via noise transition matrix estimation. However, they assume that accurate noise rates are known in prior (Patrini et al., 2017), which are empirically difficult to attain. Yao et al. (2022) extend MixUp (Zhang et al., 2018) for regression to interpolate the proximal samples in the label space to improve generalization and robustness. Thanks to its regularizing effect, it can also aid the noisy label issue.

In this work, we comprehensively explore the noisy label learning problem in regression, surpassing the scope of previous studies on several fronts. Firstly, recognizing the absence of a standardized benchmark dataset for this task, we take the initiative to curate four balanced real-world datasets. These datasets span diverse domains, encompassing age estimation (Niu et al., 2016a; Yiming et al., 2021), music production year estimation (Bertin-Mahieux et al., 2011), and clothing price predic-

tion (Kimura et al., 2021). Secondly, we conduct an empirical benchmarking exercise, evaluating the performance of thirteen baselines. These baselines are thoughtfully selected from various branches of noisy label research which are extendable to regression tasks. Lastly, recognizing the unique nature of regression, we introduce a novel metric called Error Residual Ratio (ERR). It is a simple yet effective tool for evaluating sample selection and refurbishment techniques in the context of noisy label regression. Notably, existing metrics do not adequately account for the property of regression, where labels exhibit varying degrees of noise severity.

As a novel approach to address label noise in regression, we introduce the FragSel (Fragmented Selection) framework. It is rooted in one of the fundamental characteristics of regression: the continuous and ordered correlation between the label and feature space. In other words, data points similar in the feature space are likely to have similar labels. FragSel addresses the challenge for noisy regression through the fragmentation of the label space and the consideration of neighbor relations to select clean samples. Firstly, we partition the data into smaller segments (fragments) and form pairs of the most distant fragments in the label space, resulting in what we term *contrasting fragment pairs*. To leverage the collective information from these fragments, we employ neighboring relations within both the prediction and representation spaces. This is accomplished through the design of Mixture (Jacobs et al., 1991) of neighboring fragments. Furthermore, we enhance our approach with neighborhood jittering regularization, which strengthens the selection process by improving the data coverage of each mixture. This, in turn, leads to improved agreements among neighboring fragments and serves as an effective tool for mitigating overfitting.

Finally, the contributions of this work can be summarized as follows.

I. Our empirical investigation into the realm of noisy-labeled regression stands as the most comprehensive endeavor to date. In pursuit of this study, we carefully assemble four well-balanced noisy regression benchmarks by drawing from datasets of AFAD, IMDB-Clean, SHIFT15M, and MSD. We also evaluate thirteen baselines to tackle noisy label regression.

II. We present a novel framework termed FragSel (Fragmented Selection) for noisy labeled regression. FragSel leverages the inherent orderly relationships within the label and feature spaces by employing contrastive fragmentations and constructs a mixture model based on neighborhood agreements. This is further enhanced by our neighborhood jittering regularization.

III. We propose a metric termed ERR (Error Residual Ratio), specifically designed for evaluating selected or refurbished samples. ERR takes into account the diverse degrees of noise severity present within the regression labels, offering a more comprehensive and nuanced assessment.

IV. Our experiments affirm the substantial superiority of FragSel over numerous state-of-the-art noisy label learning baselines that are applicable to regression tasks.

## 2 FRAGSEL: FRAGMENTED SELECTION

In the noisy label regression problem, we are presented with a dataset denoted as $\mathcal{D} = \{\mathcal{X}, Y\}$, where each pair $(x, y)$ represents an individual sample. Here, $x \in \mathbb{R}^d$ is the input, and $y \in \mathbb{R}$ is the observed noisy label, while its ground-truth label is denoted as $y^{\text{gt}}$. The primary objective of FragSel is to sample a *clean* subset of the data, denoted as $\mathcal{S} \subset \mathcal{D}$. By training on this selected subset, we aim to enhance the overall performance of the regression model.

Fig. 3(a) overviews the FragSel framework. Initially, we divide the dataset into what we refer to as *contrasting fragment pairs* (§ 2.1), which are collectively used for the training of feature extractors (§ 2.2). Then, we employ a probabilistic approach to select samples from the dataset $\mathcal{D}$ based on neighborhood agreements, utilizing a fragment-based mixture model (§ 2.3). Finally, the regression task is performed by training on the selected data subset $\mathcal{S}$. Moreover, neighborhood Jittering can regularize the regression training for improved performance (§ 2.4). It is important to note that FragSel operates without the need for preliminary noise rate approximations, making it noise rate-agnostic. The only hyperparameters of the framework are the number of fragments denoted as $F$, the parameter $K$ used for KNN-based prediction, and the amount of jittering applied for regularization (§ 2.4).

### 2.1 FRAGMENTATION

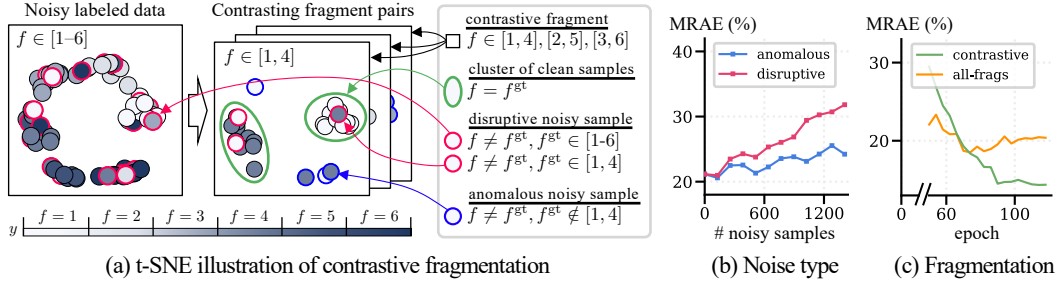

(a) t-SNE illustration of contrastive fragmentation     (b) Noise type     (c) Fragmentation

Figure 1: (a) **t-SNE illustration of contrastive fragmentation**. The data with label noise are grouped into six fragments ($f \in$ [1-6]) and formed into three contrasting pairs ($f \in [1, 4], [2, 5], [3, 6]$), leading to the pairing of distinct features. $f$ and $f^{\text{gt}}$ denote the fragment ids derived from the discretization of the continuous noisy labels and the ground truth labels, respectively. Prior to contrastive fragmentation, noisy labeled data ($f \neq f^{\text{gt}}$) are *disruptive* as they are located in the feature spaces of incorrect classes within the group ($f^{\text{gt}} \in$ [1-6]). After contrastive fragmentation, a large portion of these noisy labeled data are identified as *anomalous*, since they reside in an out-of-distribution feature space ($f^{\text{gt}} \notin [1, 4]$ while $f \in [1, 4]$). (b) When we inject disruptive and anomalous noise into a clean dataset, the disruptive ones lead to much higher errors (MRAE) in the downstream regression. (c) To select clean samples, contrastive pairings ($[1, 4], [2, 5], [3, 6]$) are more effective than using all-fragments ([1-6]), resulting in much lower MRAE scores. All experiments are based on IMDB-Clean-B with detailed settings in Appendix F.4,F.5.

We start from an inherent property of regression: data points with similar features tend to exhibit similar label values, as acknowledged in prior studies (Gong et al., 2022; Yang et al., 2022b; Yao et al., 2022). We harness this property by partitioning the continuous label space into multiple regions and organize these fragments in pairs, to maximize the distances between them (optimal in pairs of two). This affords us two key advantages for *enhanced sample selection*. Firstly, by pairing contrasting features, we can learn more discernible features, which is substantiated through the t-SNE illustration presented in Fig. 1(a) and quantitatively validated in Fig. 1(c). Notably, the model trained using contrastively paired fragments ($[1, 4], [2, 5], [3, 6]$) outperforms the one trained with all fragments ([1-6]) in terms of Mean Relative Absolute Error (MRAE). Secondly, as depicted in Fig. 1(a), noisy labeled samples are disruptive as they are used for training with wrong labels (e.g. $f \neq f^{\text{gt}}$ and $f, f^{\text{gt}} \in$ [1-6]).

1. Discretize the continuous label dataset (e.g., label range: $Y \in [10, 70]$)

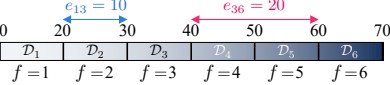

2. Graphical view of fragmented datasets

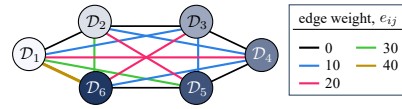

3-4. Find the best perfect matching $\mathcal{P}$ with the largest minimal edge weight

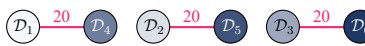

Figure 2: Contrastive Fragmentation Algorithm.

However, after contrative fragmentation, many noisy labeled samples are transformed into anomalous ones, residing in an out-of-distribution feature space (e.g. $f^{\text{gt}} \notin [1, 4]$ while $f \in [1, 4]$), which can be easily ignored during sample selection. Fig. 1(b) provides further evidence, illustrating that disruptive samples have a considerably more adverse impact on learning compared to anomalous samples, as reflected in the MRAE scores.

**Contrastive Fragmentation Algorithm**. The procedure to obtain the maximally contrasting fragment pairs is described below with an illustration in Fig. 2. The decision to utilize fragments of equal lengths and select edge weights based on the label distance between the nearest samples of fragments is motivated by our aim to fully capitalize on the advantages of contrastiveness (Shawe-Taylor & Cristianini, 1998; Grønlund et al., 2019; 2020):

1. We first divide the range of continuous labels $Y$ into an $F$ even number of equal-length fragments. As a result, we can divide the dataset $\mathcal{D}$ into an $F$ number of disjoint subsets: $\mathcal{D} = \{\mathcal{D}_1, ..., \mathcal{D}_F\}$, where each $\mathcal{D}_i$ contains the data samples whose $y$ values are in the $i$-th fragment label range.

2. We construct a complete graph $g := \{\mathcal{D}, E\}$, where each vertex denotes a data fragment $\mathcal{D}_i$, and each edge weight $e_{ij}$ is the distance in the label space between the closest samples of the fragments ($\mathcal{D}_i, \mathcal{D}_j$).

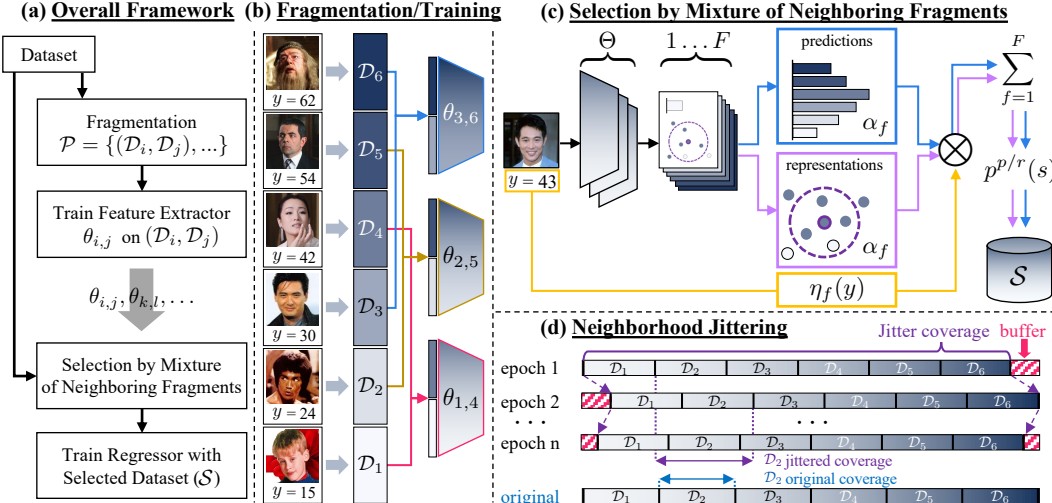

Figure 3: **Fragmented Selection framework.** (a) The overall sequential process of our framework. (b) Shows the fragmentation of the continuous label space (§ 2.1) to obtain *contrasting fragment pairs* and train feature extractors on them (§ 2.2). (c) Sample Selection by Mixture of Neighboring Fragments obtains the selection probability in both prediction and representation perspectives (§ 2.3). (d) Illustration of Neighborhood Jittering (§ 2.4).

3. We find all possible *perfect matchings* (Monfared & Mallik, 2016; Gibbons, 1985), where every vertex of a graph is incident to exactly one edge in the graph.

4. We find the perfect matching with the largest minimal edge weight: $\mathcal{P} = \arg\max_{\bar{g} \in \mathcal{G}} \left( \min \nu(\bar{g}) \right)$, where each $\bar{g}$ is a perfect matching (graph), and $\nu(\bar{g})$ is the set of edge weights in $\bar{g}$. Finally, we obtain maximally *contrasting pairs* of fragments in $\mathcal{P} = \{(\mathcal{D}_i, \mathcal{D}_j), \ldots, (\mathcal{D}_k, \mathcal{D}_l)\}$.

## 2.2 TRAINING OF FEATURE EXTRACTORS FOR CONTRASTIVE PAIRS

Once we obtain $\mathcal{P}$, as depicted in Fig. 3(b), we train a series of feature extractors, denoted as $p(y|x; \theta_{i,j})$, with parameters $\theta_{i,j}$ for every contrastive pair $(\mathcal{D}_i, \mathcal{D}_j) \in \mathcal{P}$. We have a total of $F/2$ feature extractors, which play a crucial role in generating embeddings and predictive features for each fragment. While the choice of the loss for the feature extractors can vary, we investigate both regression-based and discriminative losses in our experiments. However, we ultimately opt for discriminative training of the feature extractors due to its superior empirical performance. In the discriminative approach, we train a discriminator $p(f|x; \theta_{i,j})$, a binary classifier using the contrastive fragment ids, denoted as $f \in \{1, \ldots, F\}$, as labels.

## 2.3 MIXTURE OF NEIGHBORING FRAGMENTS

As illustrated in Fig. 3(c), the next step is to perform sample selection, whose key concept is to achieve neighborhood agreement. This is grounded in the idea that data points sharing similar features are likely to be located in proximity within the continuous label space. To deem a sample as clean, it is imperative that the fragments within the neighborhood (**Neighborhood Prior**) exhibit a consensus response (**Neighborhood Agreeability**). This consensus is evaluated based on the features derived from their respective training via contrastive fragment pair in § 2.2. Furthermore, the impact of neighbor relations is magnified by considering both representation and predictive features.

The neighborhood-based selection is formulated by a Mixture of Experts (Jacobs et al., 1991) that collectively models the contrasting fragments. The sampling probability of data $(x, y)$ is defined by

$$p(s|x, y, \mathcal{D}_{1\ldots F}; \Theta) = \sum_{f}^{F} \eta_f(y)\alpha_f(x; \mathcal{D}_{1\ldots F}, \Theta), \qquad (1)$$

where $\Theta$ denotes all parameters of $|\mathcal{P}|$ trained feature extractors from § 2.2. $\eta_f$ is the mixture weight defined via neighborhood prior, and $\alpha_f$ is the neighborhood agreeability.

**Neighborhood Prior**. For a sample $(x, y)$, we define its neighborhood prior $\eta_f(y)$ with respect to each fragment $f$. More precisely, it is determined through a softmax weighting of each fragment, taking into account its relative distance to $y$:

$$\eta_f(y) = \frac{\exp(g_f(y))}{\sum_{f'}^{F} \exp(g_{f'}(y))}, \quad \text{where } g_f(y) = \frac{\max(Y) - \min(Y)}{|y - \bar{Y}_f|}. \tag{2}$$

$g_f(y)$ decreases with a growing distance between $y$ and $\bar{Y}_f$, the mean label value of fragment $f$. $\max(Y) - \min(Y)$ is the label range, which is constant for a given dataset. Consequently, $\eta_f(y)$ rapidly decreases when the fragment $f$ is located far from $y$ in the continuous label space.

**Neighborhood Agreeability**. Since training with noisy labels leads to poorly calibrated outputs (Bae et al., 2022; Wu et al., 2020a; Zhou et al., 2021), we introduce the notion of neighborhood agreement from both predictive and representational aspects to guide sample selection. For the predictive aspect, we utilize the softmax output likelihood, while for the representational aspect, we consider the count of identical labels (fragment id, $f$) among the $k$-nearest neighbor features. We collectively denote both aspects as $\text{score}(f|x; \theta_{f,f^+}, \mathcal{D}_{f,f^+})$, where $f^+$ represents the contrasting pair for $f$, and $\mathcal{D}_{f,f^+}$ and $\theta_{f,f^+}$ refer to the data and model parameters for $f$ and $f^+$. Subsequently, the neighborhood-agreement of fragment $f$ entails self-agreement ($\alpha_f^{\text{self}}$) and its neighbor-agreement ($\alpha_f^{\text{ngb}}$) on the immediate left or right ($\alpha_{f_L}^{\text{self}}, \alpha_{f_R}^{\text{self}}$) [1]. Each agreement simply checks that the corresponding fragment $f$ outputs the largest score within its respective mixture. This is formally defined as,

$$\text{score}(f; x, \theta_{f,f^+}, \mathcal{D}_{f,f^+}) = \begin{cases} \frac{\exp(h_D(f|x; \theta_{f,f^+}))}{\sum_{f'}^{f,f^+} \exp(h_D(f'|x; \theta_{f,f^+}))} & \text{for prediction} \\ \sum_{k'}^{K_x} [f = k'] & \text{for representation} \end{cases} \tag{3}$$

$$\alpha_f^{\text{self}} = \left[ \text{score}(f; x, \theta_{f,f^+}, \mathcal{D}_{f,f^+}) > \text{score}(f^+; x, \theta_{f,f^+}, \mathcal{D}_{f,f^+}) \right] \tag{4}$$

$$\alpha_f^{\text{ngb}} = \left[ \alpha_{f_L}^{\text{self}} \vee \alpha_{f_R}^{\text{self}} \right] \tag{5}$$

$$\alpha_f(x; \mathcal{D}_{1...F}, \Theta) = \alpha_f^{\text{self}} \cdot \alpha_f^{\text{ngb}} \tag{6}$$

where $h_D(\cdot)$ is the discriminator output, $K_x$ is the label list (fragment ids) of $k$-nearest neighbors of $x$ in the representation space, $[A]$ is the Iverson bracket where $[A] = 1$ if $A$ is true, and 0 otherwise.

The predictive inference output is a scalar value when using a regression feature extractor. In that case, $\text{score}(f; x)$ is defined using distances to the contrasting pair, $f, f^+$,

$$\text{score}(f; x, \theta_{f,f^+}, \mathcal{D}_{f,f^+}) = -|\bar{Y}_f - h_R(x; \theta_{f,f^+})| \tag{7}$$

where $\bar{Y}_f$ is the average of the $f$-th fragment's labels, and $h_R(\cdot)$ is the regression function output.

By considering the neighborhoods and their agreements in predictive or representational inference outputs, we compute their corresponding sample probabilities, denoted as $p^p(\cdot)$ and $p^r(\cdot)$. Subsequently, $\mathcal{S}^p$ and $\mathcal{S}^r$ are probabilistically sampled using $p^p(\cdot)$ and $p^r(\cdot)$ respectively, and are then combined as $\mathcal{S} = \mathcal{S}^p \cup \mathcal{S}^r$.

## 2.4 NEIGHBORHOOD JITTERING

A potential limitation of mixture models is that the individual models may not fully benefit from the synergistic effect of the full dataset as they model the data from disjoint subsets (Dukler et al., 2023). Our neighborhood jittering mitigates this limitation while providing robust regularization by expanding the effective coverage of each contrastive fragment pair during learning. The detailed process is visualized in Fig. 3(d). Specifically, we bound the ratio of buffer range to jitter within as $\left[0, \frac{1}{2(F-1)}\right]$, where $F$ is the fragment number. Then, for every epoch, we shift the dataset label coverage by the randomly sampled value from the buffer range. Jittering leads to a partially overlapping mixture model (Heller & Ghahramani, 2007b; Hinton, 2002) as increasing the effective coverage per mixture allows modeling points that belong to multiple mixtures *i.e.*, neighboring fragments. Given

---

[1]We consider only a single neighbor for the right/left-most fragments in the label space.

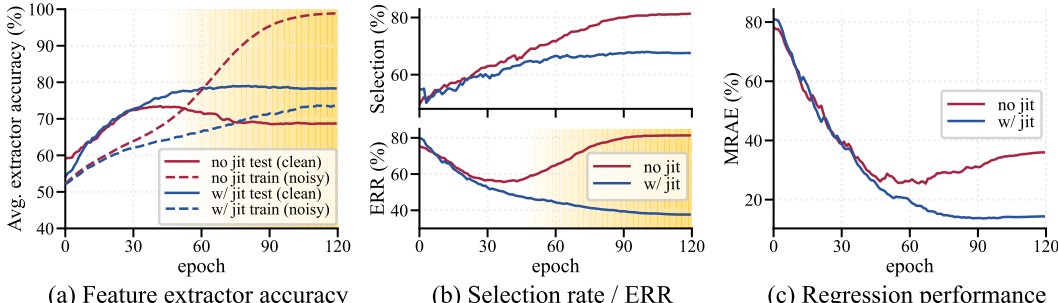

(a) Feature extractor accuracy      (b) Selection rate / ERR      (c) Regression performance

Figure 4: **Jittering analysis.** (a) Trained without jittering, feature extractors easily overfit the noisy training data (yellow-shaded region) while jittering-regularized feature extractors robustly learn from the noisy training data. (b) Overfitted feature extractors (yellow-shaded region) on noisy samples increase their likelihood, leading to a higher selection rate and ERR. It exhibits nearly twice higher ERRs (a lower value is better). (c) Most importantly, jittering regularization improves performance in regression. The analysis is done on IMDB-Clean-B with symmetric 40% noise, both with and without an additional 5% buffer range jittering.

that FragSel's effectiveness hinges on neighbor agreements, the jittering-induced overlap in training naturally enhances the learning of neighboring fragments, which is pivotal for the sample selection process during Neighborhood Agreeability (§ 2.3). Fig. 4(a) shows that when jittering is applied, the feature extractor exhibits relatively higher accuracy on the clean test data due to its regularization effect. In the sample selection stage (Fig. 4(b)), the feature extractor trained without jittering easily overfits the noise and increases the likelihood of them, resulting in over-selection and higher ERR. In contrast, the jittered (regularized) feature extractor achieves the optimal selection rate (around 60%) with half the ERR. Lastly, as shown in Fig. 4(c), the inclusion of jittering ultimately allows us to achieve significantly better performance in regression. In Appendix F.3, F.9, we provide the amount of jittering analysis and a comparative analysis of jittering to other regularization techniques, demonstrating its efficacy.

## 3 RELATED WORKS

Below, we discuss the related works on learning with noisy labels but also include a comprehensive survey in Appendix D.1. As vibrant as the research is in learning with noisy labels, there are multiple research directions. We organize it into those exploring the representations, predictions, and combination of the two.

**Prediction-based** methods have been the focus of much existing research, and they can cover a wide array of topics. There are works grounded on the small loss selection by exploring the pattern of memorization in neural networks (Han et al., 2018; Arazo et al., 2019), relying on the consistency of predictions to select or refurbish the samples (Liu et al., 2020; Huang et al., 2020), estimating the noise distribution to aid the learning (Patrini et al., 2017; Hendrycks et al., 2018), introducing an auxiliary parameter or label (Pleiss et al., 2020; Hu et al., 2020), using unlabeled data with semi-supervised learning (Li et al., 2020a; Bai et al., 2021; Karim et al., 2022), as well as designing a noise-robust loss function (Menon et al., 2020; Wang et al., 2019).

**Representation-based** methods have seen a recent surge in interest. They include selection based on clustering (Mirzasoleiman et al., 2020; Wu et al., 2020b), feature eigen-decomposition to filter (Kim et al., 2021), obtaining neighbor information to sample and also refurbish with a clean validation (Li et al., 2022a; Gao et al., 2016), and lastly, employing a generative model of features to sample (Lee et al., 2019). Some works have also studied the combination of representation and prediction spaces. For instance, Wang et al. (2022) formulate a penalized regression between the network features and the labels for selection, and Ma et al. (2018) use intrinsic dimensionality and consistent predictions to refurbish. and Ma et al. (2018) use intrinsic dimensionality and consistent predictions to refurbish. Moreover, Wu et al. (2021) expands the scope of the noisy label problem to a broader open-world scenario, and addresses it through a noisy graph cleaning framework. In accordance, our approach simultaneously employs the agreement of neighbors in the prediction and representation spaces to perform sample selection. Other important approaches include regularization via MixUp (Zhang et al.,

2018) along with its regression version (Yao et al., 2022), model-based methods that discourage large parameter shifts (Hu et al., 2020), as well as importance discrimination of parameter updates (Xia et al., 2021).

The majority of previous works address noisy labels for classification. Hence, a large portion of these works may not be directly applicable to the regression task due to the restricted usage of class-wise information. In § 4, we list some works that can be expanded to the regression task with some or minor technical adaptation.

## 4 EXPERIMENTS

We compare FragSel against thirteen powerful baselines adapted for noisy label regression. Due to the scarcity of benchmark datasets for this task, we update existing datasets to facilitate the focused study of noisy labels in the continuous space. Furthermore, to provide a novel perspective on the assessment of selection and refurbishment approaches, we introduce a new metric termed Error Residual Ratio (ERR). Lastly, we analyze our FragSel approach from many aspects to gain insights from various angles.

### 4.1 SETTINGS

**Curation of Benchmark Datasets.** We create four benchmark datasets for noisy labeled regression to encompass a sufficient quantity of data when balanced, span multiple domains, and present a significant level of complexity to pose a meaningful challenge. (1) *Age Prediction* is a well-studied regression problems (Li et al., 2019; Shin et al., 2022; Lim et al., 2020). To address this domain, we acquire two prominent datasets, **AFAD** (Niu et al., 2016a) and **IMDB-Clean** (Rothe et al., 2018; Yiming et al., 2021). To ensure fair comparisons, a ResNet-50 backbone is used across all regression tasks. (2) *Commodity Price Prediction* is a vital real-world task (Wen-Huang et al., 2021); we opt for the **SHIFT15M** dataset (Kimura et al., 2021) due to the diversity and scale of this domain. This dataset is provided as the penultimate feature of the ImageNet pretrained VGG-16 model. Consequently, all experiments use a three-layer MLP architecture (Papadopoulos et al., 2022; Kimura et al., 2021). (3) *Music Production Year Estimation* uses the tabular **MSD** dataset (Bertin-Mahieux et al., 2011). Notably, this dataset is identified as one of the most intricate and challenging datasets, based on the test R2 score (Grinsztajn et al., 2022). For all regression tasks, we adopt a tabular ResNet proposed by Gorishniy et al. (2021). To focus our investigation on the noisy label problem, we take measures to balance the datasets, a process elaborated in Appendix E.1 along with the training settings.

**Experimental Design.** We inject symmetric and Gaussian noise into the dataset labels, as done in prior literature on label noise (Yao et al., 2022; Yi & Wu, 2019; Wei et al., 2020). They can simulate a low-cost (human expert-free) controlled setting in real-world scenarios. Symmetric noise simulates the randomness such as Web crawling or annotator errors. Gaussian noise simulates the assumption that regression label noise is often Gaussian distributed around its ground-truth label. Specifically, Yao et al. (2022) inject a *fixed* 30% standard deviated Gaussian noise for *every label*, but we make it more realistic by *randomizing* the standard deviation up to 30% or 50% of the given domain's range. FragSel experiments assume the simplest setting by fixing the fragment number ($F$) as four.

**Baselines.** There exist many branches of noisy labeled learning for classification. To study the noisy label regression task, we assess thirteen baselines from three branches that are naturally adaptable to regression with minor or no update. (i) Small loss / Selection: CNLCU-S,H (Xia et al., 2022), Sigua (Han et al., 2020), SPR (Wang et al., 2022), BMM (Arazo et al., 2019), DY-S (Arazo et al., 2019). (ii) Regularization: C-mixup (Yao et al., 2022), RDI (Hu et al., 2020), CDR (Xia et al., 2021), D2L (Ma et al., 2018). (iii) Refurbish: AUX (Hu et al., 2020), Selfie (Song et al., 2019), Co-Selfie (Song et al., 2019). Comprehensive details of the baselines are in Appendix E.2.

### 4.2 EVALUATION METRICS

We mainly report the Mean Relative Absolute Error (MRAE) for all experiments. We also report the Selection rate and the error residual ratio (ERR) for selection/refurbish-based approaches. The MRAE is computed as $(e/\rho) - 1$, where $e$ is the model's MAE performance under varying conditions

Table 1: **Mean Relative Absolute Error** to the noise-free Vanilla model on the AFAD-B, IMDB-Clean-B, SHIFT15M-B, MSD-B dataset. Lower is better. A negative value indicates it performs even better than the noise-free trained Vanilla model. The results are the mean of three random seed experiments. The best and the second best methods are respectively marked in red and blue. FragSel-R,-D refers to regression-based and classification-based feature extractors, respectively. CNLCU-S/H, Co-Selfie, and Co-FragSel use dual networks to teach each other as done in Han et al. (2018). SPR (Wang et al., 2022) fails to run for SHIFT15M-B due to excessive memory consumption.

| | AFAD-B | | | | | | IMDB-Clean-B | | | | | |
| | symmetric | | | | Gaussian | | symmetric | | | | Gaussian | |
| noise rate | 20 | 40 | 60 | 80 | 30 | 50 | 20 | 40 | 60 | 80 | 30 | 50 |
|---|---|---|---|---|---|---|---|---|---|---|---|---|
| Vanilla | 9.37 | 20.27 | 30.65 | 43.09 | 28.77 | 39.03 | 16.18 | 32.05 | 53.13 | 76.35 | 26.89 | 50.28 |
| CNLCU-S | 10.98 | 20.44 | 32.44 | 41.99 | 30.60 | 40.66 | 51.40 | 66.62 | 82.83 | 85.65 | 83.39 | 82.10 |
| CNLCU-H | 4.63 | 16.32 | 36.01 | 44.71 | 35.68 | 43.64 | 6.84 | 31.16 | 63.08 | 82.65 | 46.53 | 65.24 |
| Sigua | 5.96 | 21.09 | 43.33 | 49.71 | 42.52 | 46.19 | 9.82 | 46.17 | 77.59 | 85.62 | 60.97 | 77.42 |
| SPR | 9.74 | 18.85 | 30.43 | 43.25 | 28.50 | 39.69 | 14.47 | 32.44 | 54.88 | 79.37 | 25.67 | 51.05 |
| BMM | 5.60 | 15.00 | 39.15 | 46.41 | 30.96 | 44.00 | 8.85 | 21.54 | 55.57 | 80.40 | 24.33 | 57.21 |
| DY-S | 6.87 | 15.56 | 32.24 | 45.72 | 24.40 | 43.41 | 10.42 | 21.90 | 49.94 | 78.16 | 24.70 | 44.56 |
| C-Mixup | 2.74 | 14.80 | 27.17 | 41.95 | 24.28 | 36.91 | 8.82 | 27.74 | 50.87 | 76.79 | 21.92 | 47.04 |
| RDI | 10.64 | 21.80 | 39.32 | 47.07 | 37.33 | 44.41 | 16.35 | 29.33 | 55.91 | 79.92 | 25.69 | 51.35 |
| CDR | 10.26 | 18.71 | 32.27 | 43.38 | 29.74 | 39.21 | 17.47 | 32.19 | 54.75 | 75.45 | 28.46 | 51.73 |
| D2L | 9.43 | 20.75 | 31.25 | 44.50 | 28.86 | 40.10 | 16.94 | 33.85 | 55.54 | 76.28 | 29.30 | 52.44 |
| AUX | 6.15 | 19.01 | 31.16 | 42.83 | 28.28 | 39.05 | 12.58 | 28.82 | 52.33 | 76.75 | 23.27 | 49.42 |
| Selfie | 16.91 | 25.02 | 44.18 | 47.78 | 46.02 | 50.73 | 27.43 | 53.74 | 79.38 | 84.00 | 60.68 | 78.03 |
| Co-Selfie | 14.61 | 22.95 | 39.79 | 47.72 | 41.05 | 53.00 | 23.52 | 50.07 | 67.42 | 84.25 | 52.44 | 74.73 |
| Superloss | 7.36 | 18.24 | 29.78 | 44.26 | 27.59 | 42.96 | 8.97 | 22.70 | 45.77 | 75.11 | 23.28 | 48.83 |
| **FragSel-R** | 4.97 | 13.93 | 27.85 | 37.19 | 21.93 | 33.90 | 8.74 | 22.73 | 44.29 | 68.14 | 21.74 | 46.93 |
| **Co-FragSel-R** | 2.23 | 10.22 | 22.55 | 37.55 | 21.87 | 33.73 | 2.61 | 16.06 | 40.21 | 68.00 | 18.49 | 48.79 |
| **FragSel-D** | 2.74 | 8.16 | 15.91 | 34.42 | 17.49 | 27.31 | 5.08 | 12.64 | 27.26 | 61.24 | 15.70 | 33.36 |
| **Co-FragSel-D** | 0.54 | 7.25 | 16.65 | 33.93 | 17.43 | 28.26 | 1.50 | 9.45 | 28.44 | 61.36 | 14.87 | 35.88 |

| | SHIFT15M-B | | | | | | MSD-B | | | | | |
| | symmetric | | | | Gaussian | | symmetric | | | | Gaussian | |
| noise rate | 20 | 40 | 60 | 80 | 30 | 50 | 20 | 40 | 60 | 80 | 30 | 50 |
|---|---|---|---|---|---|---|---|---|---|---|---|---|
| Vanilla | 9.11 | 17.96 | 27.02 | 36.34 | 6.54 | 15.16 | 8.23 | 18.43 | 31.67 | 45.85 | 6.96 | 15.74 |
| CNLCU-S | 12.98 | 19.42 | 24.31 | 34.47 | 15.33 | 20.90 | 0.13 | 6.04 | 21.52 | 46.01 | 4.75 | 12.51 |
| CNLCU-H | 6.26 | 12.84 | 20.04 | 36.03 | 8.88 | 15.65 | 0.27 | 4.98 | 10.32 | 29.83 | 5.11 | 9.22 |
| Sigua | 6.94 | 14.09 | 26.08 | 37.03 | 10.32 | 17.44 | 1.29 | 7.19 | 17.35 | 50.87 | 6.80 | 12.38 |
| SPR | - | - | - | - | - | - | 7.07 | 18.19 | 33.39 | 45.61 | 5.01 | 15.36 |
| BMM | 6.96 | 12.42 | 18.64 | 26.79 | 7.58 | 13.13 | 3.32 | 10.30 | 23.40 | 43.56 | 5.29 | 11.85 |
| DY-S | 7.11 | 11.94 | 18.85 | 29.04 | 6.90 | 13.50 | 3.39 | 8.06 | 18.65 | 35.24 | 4.77 | 9.83 |
| C-Mixup | 9.47 | 16.15 | 24.08 | 34.17 | 5.88 | 14.51 | 3.75 | 13.13 | 26.73 | 40.90 | 2.96 | 10.97 |
| RDI | 9.91 | 17.92 | 26.63 | 36.29 | 7.08 | 15.18 | 21.04 | 30.09 | 38.78 | 49.49 | 19.19 | 27.88 |
| CDR | 9.52 | 17.78 | 26.97 | 35.97 | 7.14 | 15.17 | 7.83 | 17.86 | 32.83 | 45.91 | 6.73 | 16.92 |
| D2L | 9.25 | 18.03 | 26.55 | 36.23 | 6.34 | 15.60 | 7.13 | 19.96 | 32.47 | 46.64 | 5.51 | 15.54 |
| AUX | 7.74 | 16.95 | 26.61 | 36.47 | 4.92 | 14.40 | 6.12 | 18.18 | 31.09 | 45.70 | 5.21 | 15.45 |
| Selfie | 4.84 | 10.22 | 22.28 | 38.15 | 5.51 | 11.58 | 1.43 | 8.40 | 20.24 | 45.87 | 14.37 | 24.13 |
| Co-Selfie | 11.53 | 16.43 | 32.08 | 39.32 | 13.45 | 22.33 | -0.38 | 4.41 | 8.32 | 35.47 | 6.78 | 13.15 |
| Superloss | 5.44 | 12.26 | 23.23 | 35.24 | 5.60 | 13.28 | 7.61 | 8.57 | 10.18 | 12.23 | 8.61 | 10.39 |
| **FragSel-R** | 4.18 | 9.59 | 16.21 | 25.76 | 4.96 | 10.90 | 0.77 | 5.68 | 13.63 | 30.05 | 2.79 | 6.87 |
| **Co-FragSel-R** | 1.82 | 7.67 | 14.11 | 24.11 | 3.90 | 9.64 | -0.31 | 3.40 | 10.31 | 26.24 | 2.18 | 6.87 |
| **FragSel-D** | 2.46 | 6.18 | 10.68 | 19.04 | 3.66 | 8.09 | 0.57 | 4.94 | 11.22 | 23.41 | 2.39 | 6.49 |
| **Co-FragSel-D** | 0.85 | 5.52 | 10.80 | 18.83 | 3.03 | 8.70 | -0.65 | 2.98 | 8.66 | 20.53 | 1.73 | 6.00 |

(data, noise type, severity) and $\rho$ is the fixed noise-free Vanilla model's MAE for the corresponding dataset. Note that we express MRAEs in percentage for better comprehensibility. Furthermore, the traditional MAE values are also reported in Appendix F.12. The Selection rate (a.k.a prevalence) is a metric often seen in noisy classification to quantify the coverage of the total dataset, $|\mathcal{S}|/|\mathcal{D}|$ where $\mathcal{S}$ is the selected set, $\mathcal{D}$ is the total dataset.

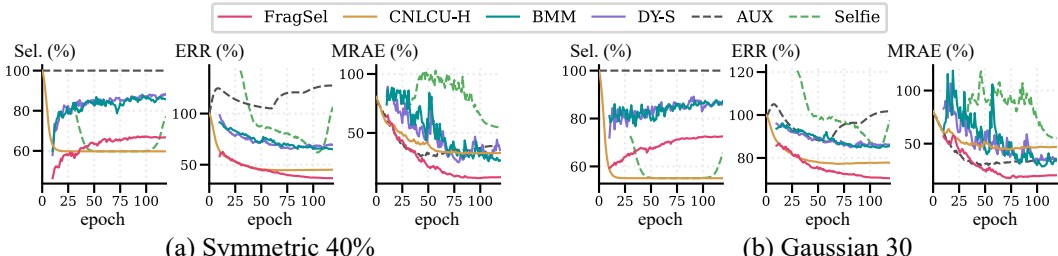

Figure 5: **Selection/ERR/MRAE comparison** between FragSel and baselines (CNLCU-H, BMM, DY-S, AUX and Selfie) on IMDB-Clean-B. We exclude the performance during the warm-up phase.

**ERR: Error Residual Ratio**. We propose a simple evaluation measure for selection and refurbishment approaches in noisy label regression tasks. A crucial characteristic of noisy regression labels is the variable severity of the noise present in each label ($y$), which can exhibit various degrees of deviation from the ground truth ($y^{gt}$). This cannot be addressed when using conventional metrics like precision and recall, since they tend to treat all instances of noise as equally severe. Our proposed Evaluation Metric for Regression Noise (ERR) considers the varying severity of noise while concurrently separating the assessment of selected or refurbished samples from the inherent capability of the regression model in mitigating the impact of noise. The metric is defined as

$$\text{Error Residual Ratio (ERR)} = \frac{\frac{1}{|\mathcal{S}|} \sum_s^{|S|} |y_s - y_s^{gt}|}{\frac{1}{|\mathcal{D}|} \sum_d^{|\mathcal{D}|} |y_d - y_d^{gt}|}. \tag{8}$$

Analyzing ERR along with the selection rate and regression metrics (*e.g.*, MSE, MRAE) provides a deeper insight into the model performance. Ideally, a method with a high selection rate, coupled with low ERR and favorable regression metric scores, can be deemed as closer to the upper bound.

## 4.3 ANALYSIS & DISCUSSION

**Overall performance.** Table 1 compares the mean relative absolute error (MRAE) to the noise-free trained Vanilla model between FragSel and the baselines. We evaluate six types of noise: four symmetric and two random Gaussian noises. FragSel-R, FragSel-D, and Co-FragSel-D achieve the strongest performance in all experiments compared to the thirteen baselines. Notably, Co-FragSel-D mixes co-teaching during the regression learning phase by assuming that $\mathcal{S}$ still contains 25% noise.

**Selection/ERR/MRAE comparison.** Fig. 5 compares the selection rate, ERR, and MRAE for FragSel and five selection/refurbishment baselines (CNLCU-H, BMM, DY-S, AUX, Selfie) on IMDB-Clean-B. An ideal model should exhibit a high selection rate and a low ERR. It is worth noting that the relative importance of ERR and selection rate may vary depending on the dataset and the task. Notably, FragSel achieves the lowest ERR while maintaining above-average selection rates, resulting in the best MRAE. This hints at a potential future direction for improvement, particularly in the area of refurbishment. Appendix F.10 includes all noise types with more baseline comparison results.

**Appendix** supplements the limitation B, parameter size comparison F.1, analysis for fragment number F.2, hyperparameter F.3, disruptive versus anomalous noise F.5 and variance F.11, ablation study F.7, performances of the discretized version of the baselines F.8, FragSel pseudo code 1.

## 5 CONCLUSION

To address the problem of noisy labeled regression, we introduced the Fragmented Selection framework (FragSel). The framework partitions the label space and identifies the most contrasting pairs of fragments, thereby facilitating the training of a mixture of feature extractors over contrasting fragments. This mixture is leveraged for clean sample selection based on neighborhood agreements. Extensive experiments on four datasets on three domains with different levels of symmetric and random Gaussian noise demonstrate that our framework performs superior selection and ultimately leads to a better regression performance than many other state-of-the-art models. FragSel, given its foundation in the Mixture of Experts model, exhibits linear growth in parameter size with an increase in the number of fragments. We acknowledge this as a potential avenue for future research.

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

## A  APPENDIX

The Appendix enlists the following additional materials.

## B  LIMITATION

A key limitation of FragSel lies in its foundational reliance on the Mixture of Experts (MoE) model (Jacobs et al., 1991). Specifically, integrating MoEs with deep learning introduces notable scalability challenges, both computationally and in memory usage (Zuo et al., 2021; Zoph et al., 2022; Zhang et al., 2021b). To address the memory concern, FragSel currently employs more compact feature extractors. Nevertheless, a prominent inefficiency stems from expert redundancy in MoEs' parameters (Zuo et al., 2021). Some approaches to mitigate this include distilling into sparse MoE models, employing pruning, and subsequently compressing to decrease parameter size (Kim et al., 2023; Fedus et al., 2021). There are also emerging strategies centered on parameter sharing, leveraging matrix product operators (MPO) decomposition (Gao et al., 2020; 2022) and parameter-efficient fine-tuning (Zadouri et al., 2023). Of these, we believe the avenue of parameter sharing holds special promise when combined with FragSel; the inherent positive feature correlation in regression problems amplifies the advantages of this approach.

In its current form, FragSel facilitates simultaneous training of both the feature extractors and the subsequent task, either on a per-batch or per-epoch basis. However, a wealth of research exists that could further optimize FragSel's scalability. These span from improving training efficiency (H et al., 2021; Zoph et al., 2022; Lepikhin et al., 2021; Lewis et al., 2021) to enhancing inference capabilities (Zhang et al., 2021b; Fedus et al., 2021).

## C  THEORY OF FRAGSEL

We present several theoretical justifications that enhance the performance of FragSel.

### C.1  FRAGMENTATION AND NEIGHBORHOOD JITTERING

FragSel operates by partitioning data samples into fragments and leveraging trained feature extractors for sample selection through collective modeling. We conceptualize this as a Mixture-of-Experts (MoE) model, wherein individual experts specialize in specific problem subspaces through data partitioning Yuksel et al. (2012); Masoudnia & Ebrahimpour (2014). MoEs possess theoretically advantageous properties with respect to computational scalability and reduction of output variance Yuksel et al. (2012), contributing to the enhancements observed in FragSel. It is noteworthy that since each network is trained on a distinct training set, MoE effectively mitigates concurrent failures, thereby preventing error propagation among networks and ultimately improving the generalization performance of FragSel as well Sharkey & Sharkey (1997).

Additionally, our Neighborhood Jittering leads to a Partially Overlapping Mixture Model Heller & Ghahramani (2007a), theoretically enabling the modeling of significantly richer and more intricate hidden representations by accommodating multi-cluster membership, ultimately enhancing the selection and overall performance of FragSel.

### C.2  CONTRASTIVE FRAGMENTATION-BASED NOISY LABELS TRAINING

Previously, Zheng et al. (2020) demonstrated that a binary classifier trained on noisy labels can effectively indicate the cleanliness of training data labels. Given that our methodology involves binary classification for contrasting fragment pairs, a similar property holds true with minor adjustments.

In Theorem 1 of Zheng et al. (2020), it is asserted that when the noisy classifier exhibits low confidence, the label is likely to be noisy with bounded probability. This is substantiated by examining the true conditional probability $\eta(x)$, Bayes optimal classifier, Tsybakov condition, transition probability, noisy classifier's prediction, and other factors. Our approach can follow the proof by simply substituting the clean and noisy label $y^{gt}, y$ with the clean and noisy fragment id $f^{gt}, f$, resulting in the assertion that the noisy binary classifier learned from contrastive pairing can assess the cleanliness of noisy labels.

Furthermore, even though the Tsybakov condition, which posits that the margin region near the decision boundary has a bounded volume, was assumed in Zheng et al. (2020)'s proof, the design of contrastive fragmentation can strengthen this condition. This occurs as contrastive fragmentation enforces a margin between paired fragments, creating a distinct gap in label space between them.

To elaborate briefly, consider a label space fragmented into four fragments (i.e., $F = 4$), each covering label ranges $(y_0^L, y_0^R), (y_1^L, y_1^R), (y_2^L, y_2^R), (y_3^L, y_3^R)$. Introducing symmetric noise at a rate $\sigma$ and pairing fragments $(0, 2)$ based on noisy fragment ids $f$, the data distribution of post-contrastive fragment pairing becomes $\Pr(f^{gt} = 0) = \Pr(f^{gt} = 2) = \frac{1}{2} - \frac{1}{4}\sigma$ and $\Pr(f^{gt} = 1) = \Pr(f^{gt} = 3) = \frac{1}{4}\sigma$. (Note that samples with clean fragment ids $(1, 3)$ exist because the pairing is performed based on the noisy fragment ids $f$.) Then, assuming the conditional probability of a sample $(x^{(i)}, y^{gt,(i)})$ follows the relative distance of the label to each fragment as

$$\eta(x^{(i)}) = \Pr(f^{gt} = 2 | x^{(i)}) = \frac{\max(\min(y^{gt,(i)}, y_2^L) - y_0^R, 0)}{y_2^L - y_0^R} \tag{9}$$

we can demonstrate that the Tsybakov condition is satisfied for data distributed in label space $y^{gt} \in (y_1^L, y_1^R)$. Specifically, $\Pr\left[|\eta(x) - \frac{1}{2}| \leq t\right] = \Pr\left[\left|\frac{y - y_0^R}{y_2^L - y_0^R} - \frac{1}{2}\right| \leq t\right] = \frac{\sigma}{4} \times 2t = \frac{\sigma}{2}t$ when $t < 0.5$. This supports the validity of the Tsybakov condition assumption in our approach.

## C.3 FRAGSEL-D VERSUS FRAGSEL-R

In Table 7, FragSel-D outperforms FragSel-R in all experiments. This is because FragSel-D's feature extractor is trained with a discriminative loss (Cross-Entropy), which results in more stable training than the regressive loss (Mean Squared Error) used in FragSel-R.

During the learning process, deep neural networks aim to maximize the mutual information between the learned representation, denoted as $Z$, and the target variable, denoted as $Y$. The mutual information between these two variables can be defined as $I(Z;Y) = H(Z) - H(Z|Y)$. A high value of $I(Z;Y)$ is indicative of a high marginal entropy $H(Z)$. Achieving this dual objective is accomplished by classification Boudiat et al. (2020).

However, Zhang et al. (2023) have shown that regression primarily focuses on minimizing $H(Z|Y)$ while disregarding $H(Z)$. This results in a relatively lower marginal entropy for the learned representation $Z$ and ultimately leads to performance deficits in comparison to classification.

# D  EXTENDED RELATED WORK

## D.1  CONTINUOUSLY ORDERED CORRELATION OF LABELS AND FEATURES

One distinctive characteristic of regression problems is their continuous label space, implying a high likelihood of correlation between regions within the feature and label spaces (Yang et al., 2022b; Gong et al., 2022; Zha et al., 2022).

Recent research has extensively explored these characteristics, encompassing issues such as label imbalance (Yang et al., 2022b; Gong et al., 2022), age estimation (Li et al., 2019), contrastive learning (Zha et al., 2022), and mixup regularization (Yao et al., 2022).

Yang et al. (2022b) propose label and feature distribution smoothing based on their similarity, while Gong et al. (2022) introduce a regularization term aimed at aligning the rankings of feature-space and label-space neighbors. Zha et al. (2022) employ supervised contrastive learning with a pairing technique based on label distances in mini-batches. To adapt MixUp (Zhang et al., 2018) for regression tasks, Yao et al. (2022) recommend interpolating proximal samples within the label space with a higher probability.

Ordinal regression, also known as ranking learning, pertains to predicting ordinal labels based on input data. It is noteworthy that ordinal regression methods are adaptable for regression tasks due to the inherent numerical ordering within scalar label spaces. Past studies in ordinal regression have successfully addressed various regression challenges, including facial age estimation (Niu et al., 2016b; Shin et al., 2022), monocular depth estimation (Fu et al., 2018), and credit rating (Hirk et al., 2019). Some of these methods share common characteristics with our approach, as they discretize continuous labels, effectively converting regression tasks into classification problems (Niu et al., 2016b; Fu et al., 2018; Shah et al., 2022). Within the framework of ordinal regression, Garg & Manwani (2020) propose a loss correction method by estimating the noise transition matrix.

It is important to note that among the previously mentioned methods, only Yao et al. (2022) and Garg & Manwani (2020) can effectively address noisy label regression problems without the need for additional techniques. Additionally, Wang et al. (2022) enhance the scalability of their approach by grouping dissimilar classes within the feature space. Our work considers the continuity of labels and features and their correlation in fragmenting and grouping data. This approach allows each component to learn distinguishable features and improve sample selection capabilities.

## D.2  NOISY LABEL IN OBJECT DETECTION

Due to the abundance of research on object detection tasks, with bounding box localization being a prominent example of regression tasks, we have explored the issue of noisy regression within the context of object detection. In particular, obtaining accurate annotations for object detection is a resource-intensive task, often constrained by limited time, a small number of annotators, or reliance

on machine-generated annotations. These constraints frequently result in label noise, represented as incorrect class assignments or inaccurate bounding box locations.

Various strategies have been developed to address the issue of noisy labels in object detection. To correct inaccurate bounding box locations, Li et al. (2020b) leverage the discrepancy between two classification heads with emphasizing the objectness of the region. Liu et al. (2022) generates object bags using the classifier as guidance, Mao et al. (2021) employs center-matching correction, and Schubert et al. (2023) drop instances with high region proposal loss on an instance-wise basis. In scenarios where image-level annotations are available, Gao et al. (2019) employs ensemble learning with two classification heads and a distillation head, while Shen et al. (2020) decomposes the problem into foreground and background noise, employing residual learning and bagging-mixup learning.

We also explored the possibility of applying object detection techniques to noisy labeled regression. However, our analysis revealed that these methods are not well-suited for the broader regression task. Specifically, Liu et al. (2022); Schubert et al. (2023); Mao et al. (2021) utilize region proposal networks to generate bounding box proposals. They leverage these proposals to selectively choose clean labels or re-weight the training samples. However, because this approach necessitates an auxiliary model in the proposal generation process, it cannot be directly applied in the context of regression tasks.

Additionally, Li et al. (2020b); Liu et al. (2022); Schubert et al. (2023); Gao et al. (2019) employ the object detector's classifier to update or assess the quality of bounding boxes. By evaluating the confidence or consistency of the bounding box through the classification output, this approach helps mitigate the impact of noisy labels. However, implementing a similar approach in the context of regression tasks would require the inclusion of an auxiliary co-trained task.

## D.3 Transition Matrix based Methods

Methods based on transition matrices constitute one of the primary approaches for addressing the issue of noisy labels.

Driven by the observation that the clean class posterior, denoted as $p(y^{gt}|x)$, can be inferred from the transition probability and the noisy class posterior, $p(y|x) = T(y|y^{gt})p(y^{gt}|x)$, the modification of the loss function enables the construction of a risk-consistent estimator using the estimated transition matrix (Yao et al., 2020).

There are many approaches aiming to enhance the estimation of the transition matrix. These include factorizing it into the product of two matrices by introducing an intermediate class (Yao et al., 2020), training the Bayes label transition network (Yang et al., 2022a), learning the transition matrix within a meta-learning framework (Wang et al., 2020), down-weighting less informative features based on $f$-mutual information (Zhu et al., 2022), and adopting a two-head architecture. The latter involves a noisy classifier for simultaneous transition matrix estimation and a clean classifier for statistically consistent training (Kye et al., 2022).

Moreover, Xia et al. (2020) explores the utilization of part-dependent transition matrices, combining them to approximate the instance-dependent transition matrix.

In an extended context, Li et al. (2022c) broadens the problem to include noisy multi-label learning and suggests considering label correlations.

## D.4 Combination with Contrastive Learning

Incorporating unsupervised learning methods proves effective in alleviating label noise, prompting the integration of noisy label mitigation techniques with unsupervised learning, particularly contrastive learning.

Zhang et al. (2021a) show that the combination of contrastive loss and semi-supervised loss yields successful mitigation of the noisy label problem.

Beyond the application of contrastive learning, other approaches involve selecting confidence pairs and confidence samples (Li et al., 2022b), leveraging clean probability estimation derived from

Table 2: **Dataset Statistics** on the four newly curated balanced datasets for regression. AFAD-B (Niu et al., 2016a), IMDB-Clean-B (Yiming et al., 2021), SHIFT15M-B (Kimura et al., 2021), MSD-B (Bertin-Mahieux et al., 2011).

| Dataset | range | train | valid | test | total |
|---|---|---|---|---|---|
| AFAD-B | [15, 40] | 27647 | 1627 | 3252 | 32526 |
| IMDB-Clean-B | [15, 66] | 44200 | 2600 | 5200 | 52000 |
| SHIFT15M-B | [0, 40000] | 273417 | 16080 | 32180 | 321677 |
| MSD-B | [1956, 2010] | 25218 | 1512 | 2970 | 29700 |

the relationship between representation clusters and labels (Huang et al., 2023), employing class prototypes for weakly-supervised loss (J. Li, 2021), and implementing soft-labeling based on the relation between representations and labels (Ortego et al., 2021).

Additionally, an approach introduces a contrastive regularization function aimed at preventing adverse effects stemming from noisy labels (Yi et al., 2022).

# E  EXPERIMENT DETAILS

## E.1  DATASET CURATION DETAIL

Table 2 provides a comprehensive overview of the statistics for the four benchmark datasets meticulously curated for the task of noisy label regression. Detailed descriptions of the dataset tailoring process are presented below for clarity.

**IMDB-Clean-B** and **AFAD-B**: These datasets are harmonized by achieving a balance across distinct age values. This equilibrium is established using a bin threshold (clip value) of 1000 and 1251 sample counts for IMDB-Clean-B and AFAD-B, respectively. To ensure uniformity, image inputs are resized to dimensions of $(128 \times 128)$. For the regression task, we consistently employ a ResNet-50 backbone across all models.

**SHIFT15M-B**: Achieving data balance in this dataset involves a two-step process. First, the label space is binned based on a price threshold of ¥2000. Subsequently, data points exceeding the maximum price of ¥40000 are clipped to remove outliers. The binning threshold is set at 16084 sample counts to further ensure balanced representation. To standardize the label currency, it is pegged to the U.S. dollar, referencing exchange rates from 2010 to 2020, which coincides with the period when the original clothing item data is collected. Notably, this dataset is provided as the penultimate feature of the ImageNet pretrained VGG-16 model. Consequently, we opt for a three-layer MLP architecture with a hidden layer size of [2048, 1024, 512], aligning with recommendations from Papadopoulos et al. (2022) and Kimura et al. (2021).

**MSD-B**: Achieving balance in the Million Song Dataset involves setting a threshold of 550 samples per year. For all regression models in this context, we adopt a regression backbone rooted in the tabular ResNet structure proposed by Gorishniy et al. (2021), featuring a hidden dimension of 467.

## E.2  BASELINES DETAILS

While numerous branches of noisy labeled learning have been explored for classification tasks, our focus in this study centers on the challenging domain of noisy label regression. To comprehensively investigate this task, we have conducted an extensive review of the various branches and have selected a set of thirteen baselines that are adaptable to regression. It is worth noting that C-Mixup (Yao et al., 2022) was originally proposed as a regression baseline. In the following section, we provide an overview of these selected baselines, offering a broad coverage of diverse approaches to address the noisy label regression problem. Additionally, we present detailed descriptions of the experimental settings for each baseline.

1. D2L (Ma et al., 2018) for intrinsic dimension exploration. Following the paper, we set $k = 20$ and $m = 10$ for Local Intrinsic Dimensionality (LID) estimation and set the LID estimation window as five following the official implementation.

2. CDR (Xia et al., 2021) for model weight parameter selection, and RDI (Hu et al., 2020) for regularizing the paramter distance from the initialization. At RDI, we use search space $\lambda \in [0.25, 0.5, 1, 2, 4, 8]$.

3. C-Mixup (Yao et al., 2022) to regularize via continuous mixup. C-Mixup-batch is used in all experiments because of the excessive memory requirement for pairwise distance matrix $P$. We set the beta distribution variable $\alpha$ as 1.5. The bandwidth variable $\sigma$ is searched over [0.01, 0.1, 1], following Yao et al. (2022).

4. SELFIE (Song et al., 2019) and AUX (Hu et al., 2020) for refurbishing. To apply SELFIE to the continuous label, we redefine the concept of uncertainty $F(x; q)$ and refurbished labels $y^{refurb}$ with the mean and standard deviation.

$$F(x; q) = \frac{\sigma(H_x(q))}{(\max{(Y)} - \min{(Y)})} < \epsilon \tag{10}$$

$$y^{refurb} = \mu(H_x(q)) \tag{11}$$

where $H_x(q)$ is the prediction history of $x$ from before $q$ epochs, $\epsilon$ is the uncertainty threshold.

For SELFIE, we train 1/4 of the total training epochs for the warm-up phase, following Song et al. (2019). The variable $q$ is searched over half of the warm-up epochs and around. The variable $\epsilon$ is searched over [0.05, 0.10, 0.15, 0.20], following Song et al. (2019). For Co-Selfie, we search over the same parameters as Co-FragSel.

For AUX (Hu et al., 2020), we regularize the auxiliary variable by weight decay 0.0005, reducing the weight by 0.1 at 1/2 and 3/4 of the total training epochs. The learning rate of the auxiliary variable is set to 0.1 and 0.01. The variable $\lambda$ is searched over [0.25, 0.5, 1, 2, 4, 8].

5. SPR (Wang et al., 2022) performs penalized regression for selection. It requires some adaptation to regression by ignoring the $\ell_q$ penalty as there is no longer a linearity gap between the scalar output and the final fully connected layer that require reducing. Also, we use our fragmentation splits $4, 8$ to bin the regression data for SPR's parallel optimization.

6. Sigua (Han et al., 2020) and CNLCU-S/H (Xia et al., 2022) for small loss selection. For Sigua, we use $\delta(t) \in [0.3, 0.4]$ and $\gamma = 0.01$ and set $T_k$ as 5% of the total training epochs. For CNLCU-S/H, we search $\sigma$ and $\tau_{\min}$ in [0.01, 0.1, 1, 10] and set $T_k$ as 5%.

7. BMM (Arazo et al., 2019) for selection based on beta mixture model fitting on the loss distribution. BMM does hard sampling and trains using the selected samples. DY-S is a dynamic soft loss. We implemented two versions; the first uses a convex combination as in Reed et al. (2015) $((1 - w)\tilde{y}^c - w\hat{y})^2$. Second, instead of bootstrapping, we dynamically weight the loss using the BMM probability to create a cost-sensitive loss, $(1 - w)\ell$. The $w$ is the mixture clean probability, $\hat{y}$ is the model prediction, $\tilde{y}^c$ is the assigned noisy label, and $\ell$ is the loss.

8. [Incompatible] CRUST (Mirzasoleiman et al., 2020) for clean coreset selection. It aims to select a coreset based on *class-wisely gradient clustering*. For regression, we initially viewed *all data as a single class* and proceeded with coreset selection, but the results were unsatisfactory. Therefore, we report results based only on the discretized version, demonstrating comparable performances. We select 1/2 of the total dataset as a coreset. The distance threshold in calculating clusters is searched over [1, 2, 4].

9. [Incompatible] OrdRegr (Garg & Manwani, 2020) for loss correction. Since no official implementation is provided, we implemented it with cross-entropy loss for ordinal regression. Importantly, we failed to find accurate noise rate estimation using their suggested methods. Even when considering the transition matrix with the actual noise rate, the loss correction algorithm proved ineffective in our benchmark tests.

### E.3 FRAGSEL TRAINING DETAILS

FragSel-R's regression feature extractor employs the standard Mean Squared Error (MSE) loss. Both FragSel-R and FragSel-D employ the Cosine Annealing Learning rate (Loshchilov & Hutter,

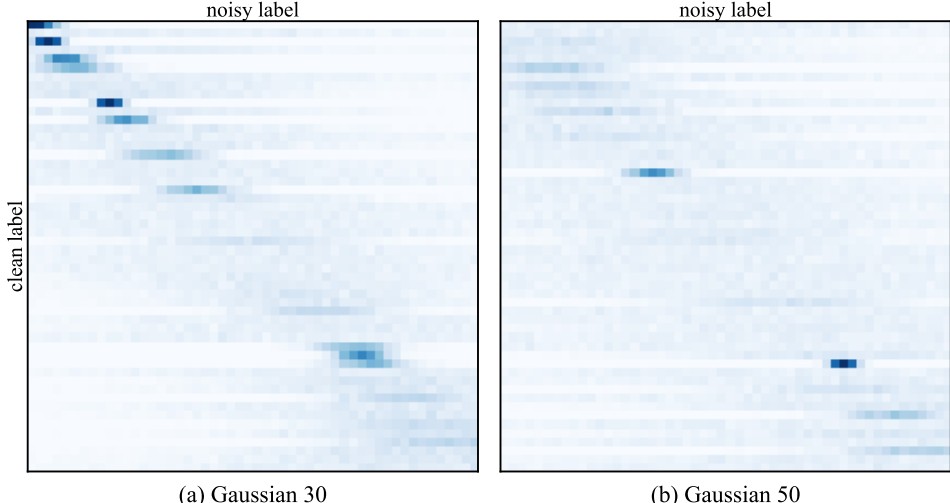

Figure 6: **Random Gaussian Noise.** (a) Gaussian noise injected from the uniformly sampled random standard deviation between $[1, 30]$. (b) Gaussian noise injected from uniformly sampled random standard deviation between $[1, 50]$.

2017) with a minimum learning rate of $\eta_{min} = 0$. The optimization is carried out using the Adam optimizer (Kingma & Ba, 2015). For the $K$-Nearest Neighbors (KNN)-based prediction, we experiment with various values of $K$, specifically choosing from the set $[3, 5, 7]$. The number of fragments, denoted as $F$, remains constant at four throughout all our experiments. To determine the buffer range for jittering, we conduct a search over values within the range $[0, 0.05, 0.1]$.

Some dataset-specific hyperparameters exist:

- Age prediction task datasets, IMDB-Clean-B (Rothe et al., 2018) and AFAD-B (Niu et al., 2016a) train for 120 epochs with learning rate of 0.001. Each feature extractor employs the ResNet-18 architecture, which contains only 48% of the parameters found in ResNet-50, the architecture utilized for the regressor.

- Clothing price estimation task dataset SHIFT15M-B (Kimura et al., 2021) trains for 40 epochs with learning rate of 0.0001. MLP with hidden dimensions [1024, 512, 256] is deployed for feature extractors, and the parameter size is 44% of the regressor's.

- Music year production task dataset MSD-B (Bertin-Mahieux et al., 2011) trains for 20 epochs with learning rate of 0.0001. Similar to the regression backbone, the feature extractor model is the tabular ResNet structureGorishniy et al. (2021), and the hidden dimension is reduced to 256.

### E.4 RANDOM GAUSSIAN NOISE

Fig. 6 illustrates the application of random Gaussian noise within the label space of IMDB-Clean-B (Rothe et al., 2018). The procedure for injecting noise is akin to the approach employed by Yao et al. (2022), where Gaussian noise is applied to every unique label within the training samples. Specifically, Yao et al. (2022) sets the standard deviation of the Gaussian noise as a fixed 30% of the label space corresponding to the dataset. In contrast, our noise injection method introduces an element of stochasticity, allowing for variable levels of deviation for each unique label.

To achieve this variability, we employ uniform sampling from the minimum and maximum values specific to each label's domain. For instance, in the context of an age prediction task, we assume minimum and maximum values of 0 and 100, respectively. However, in cases where the label domain lacks clarity (*e.g.*, for a variable like 'price'), we utilize the minimum and maximum label values provided by the dataset itself.

It is important to highlight that baselines with known noise rate, such as CNLCU-S/H, Sigua and Selfie, are incapable of dealing with Gaussian noise. Given that these baselines employ a heuristic approach to control selection rates through $(1 - \text{noise rate})$, they prove ineffective when exposed to Gaussian noise, as it introduces noise to all samples, thereby resulting in a nearly 100% noise rate. Hence, we create a *soft noise rate* to be used by them for selection. This is done by calculating an updated noise rate, assuming that the Gaussian noise injected samples that fall within an acceptable variance of the original ground-truth label are clean (the acceptable variance is set to equal the label length/size of a single fragment).

## F    EXTENDED RESULTS & ANALYSIS

We conduct supplementary experiments and analyses pertaining to parameter sizes, fragment numbers, other hyperparameters $(K, J)$, contrasting fragmentation, and the impact of disruptive or anomalous noise. Furthermore, we present ablation analyses, comparisons with discretized baselines, baseline performance evaluations considering Selection rate and ERR, variance assessments, and the obtained MAE results.

### F.1    PARAMETER SIZE COMPARISON.

Table 3 compares the number of parameters of FragSel and baselines on the ResNet-based datasets, AFAD-B and IMDB-Clean-B. A thorough description of the FragSel architecture is in Appendix E.3. It is worth noting that FragSel's feature extractors for noise mitigation employ a much fewer number of parameters than the downstream regression task. The total number of parameters in Table 3 varies, as some share parameters for regression as well as noise mitigation while others, such as FragSel, do not. Nevertheless, FragSel uses fewer total parameters than CNLCU-H and RDI.

Table 3: **Parameter size comparison**. regression: parameters for regression, noise: parameters to mitigate noisy labels, "others": SPR, CDR, D2L, C-Mixup, Sigua, Selfie, BMM, DY-S.

|  | regression | noise | total |
|---|---|---|---|
| RDI | 23.9M | 47.8M | 47.8M |
| CNLCU | 47.8M | 47.8M | 47.8M |
| "others" | 23.9M | 23.9M | 23.9M |
| FragSel-R/D | 23.9M | 22.8M | 46.7M |

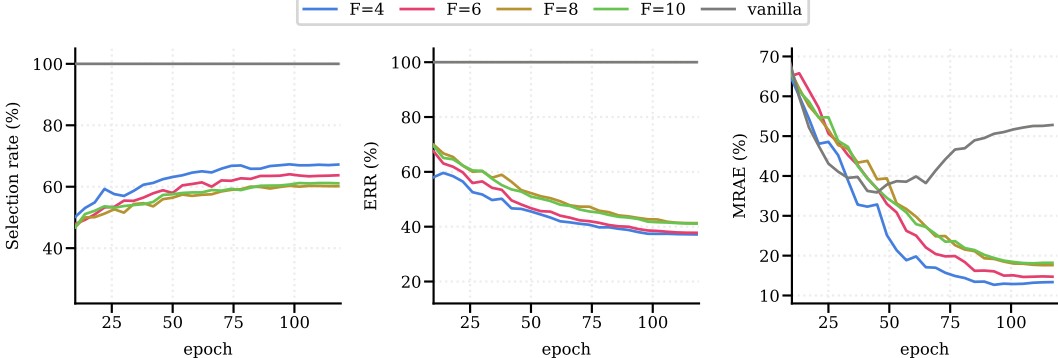

Figure 7: **Fragment number analysis** compares the Selection rate, ERR and MRAE on IMDB-Clean-B with symmetric 40% noise.

### F.2    FRAGMENT NUMBERS.

The choice of an optimal *total* number of fragments is contingent upon the dataset's inherent difficulty, an aspect that is garnering increasing attention in the research community (Ethayarajh et al., 2022). In this study, we adopt the simplest configuration by setting the total number of fragments to four, and yet, we consistently observe significant improvements in performance across all our experiments.

In Fig. 7 and 8, we undertake an examination of various fragment numbers within the context of symmetric 40% noise, using the IMDB-Clean-B and SHIFT15M-B datasets as benchmarks. Our

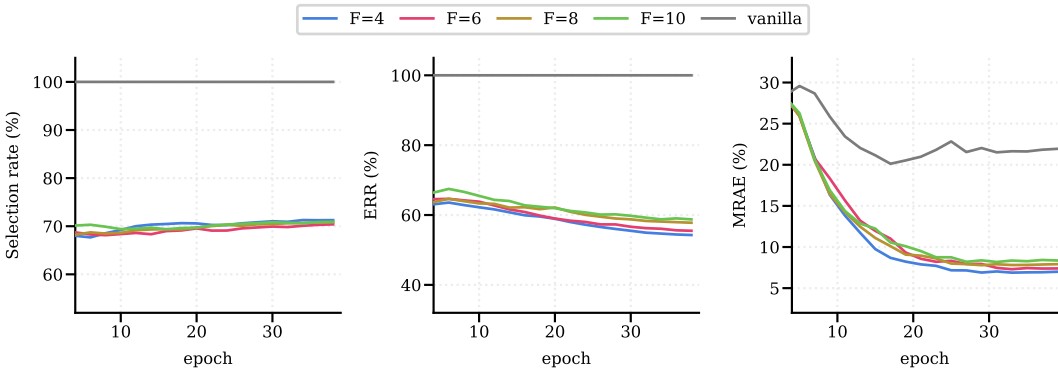

Figure 8: **Fragment number analysis** compares the Selection rate, ERR and MRAE on SHIFT15M-B with symmetric 40% noise.

evaluation criteria encompass the Selection rate, Error Residual Rate (ERR), and Mean Relative Absolute Error (MRAE). The number of fragments is chosen from $F \in [4, 6, 8, 10]$, and to address scenarios with a smaller fragment number, we examine cases where $F = 1$ or $2$. Initially, when $F = 2$, a fragment $f$ that satisfies self-agreement (Eq. 4) does not meet the criteria for neighbor-agreement (Eq. 5), as the agreement relies on comparing the scores of fragment $f$ and its contrasting pair $f^+$. Consequently, the unified neighborhood agreement (Eq. 6) consistently yields a value of $0$. On the other hand, defining a contrasting pair is not feasible when $F = 1$. As a result, the computation of the score (Eq. 3) becomes unfeasible, thereby rendering the calculation of neighborhood agreement (Eq. 6) not possible. Instead, we present a plot of the vanilla baseline to illustrate the case when $F = 1$ without utilizing FragSel.

The results reveal that the MRAE of the vanilla model initially decreases during the early epochs as it learns patterns from clean samples. However, as the model starts to memorize noisy samples, the MRAE degrades. In contrast, FragSel consistently mitigates the impact of noisy samples across all plots ($F \in [4, 6, 8, 10]$) when compared to the vanilla baseline. We also observe a declining trend in performance as the number of fragments increases in the case of IMDB-Clean-B. In contrast, SHIFT15M-B exhibits relatively stable performance across different fragment numbers. This decrease in performance with an increased number of fragments is likely attributed to a finer division of the training data among feature extractors (discriminative models), ultimately leading to reduced generalization capabilities.

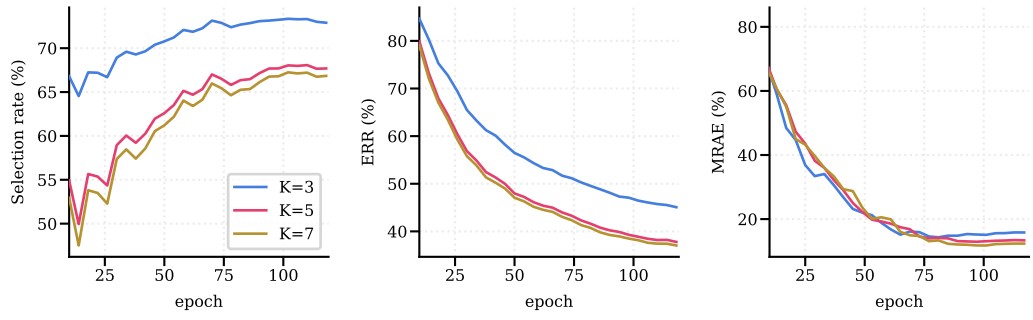

Figure 9: **Hyperparameter K analysis** compares the Selection rate, ERR and MRAE on IMDB-Clean-B with symmetric 40% noise.

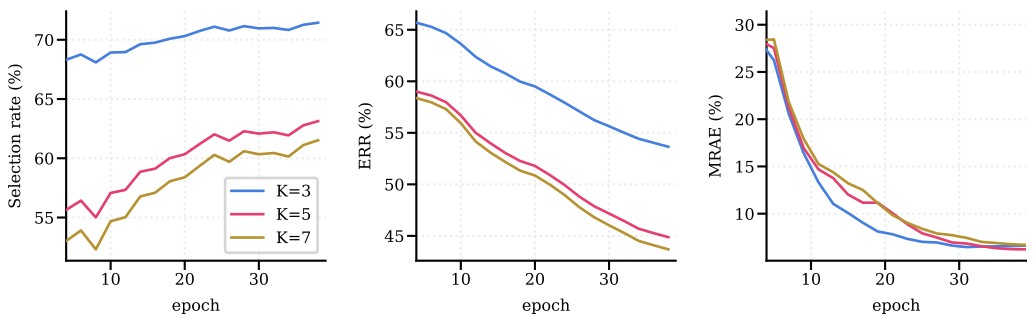

Figure 10: **Hyperparameter K analysis** compares the Selection rate, ERR and MRAE on SHIFT15M-B with symmetric 40% noise.

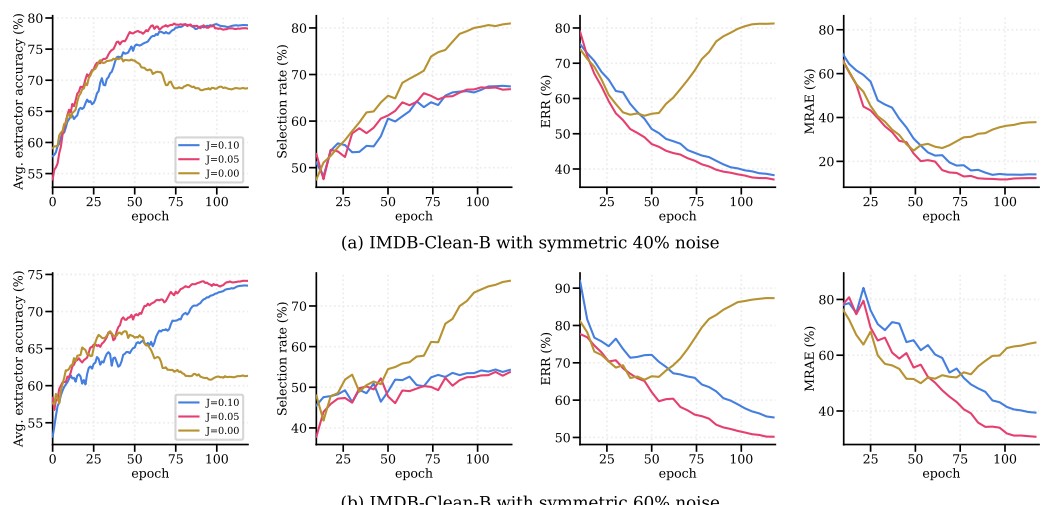

Figure 11: **Hyperparameter J analysis** compares the average accuracy of feature extractors, the Selection rate, ERR and MRAE on IMDB-Clean-B with symmetric 40%, 60% noise.

## F.3 Hyperparameter Analysis

The hyperparameter $K$ determines the number of neighbors considered when assessing self/neighbor agreement from a representation perspective. As shown in Fig. 9, 10, with an increase in the value of $K$, the criteria for agreement become more stringent. Consequently, as $K$ value increases, a greater number of confident samples are selected, resulting in a reduction in the Selection rate, ERR.

The hyperparameter $J$ controls the buffer range for jittering, which, in turn, determines the level of regularization applied via neighborhood jittering. Increasing the value of $J$ results in stronger regularization, effectively preventing overfitting. However, excessive regularization, as observed when $J = 0.10$, may result in adverse effects during training. Specifically, in Fig. 11(a), the feature extractors exhibit similar convergence patterns when $J = 0.05$ or $J = 0.10$. Consequently, comparable performance is observed in Selection Rate and MRAE. Yet, in Fig. 11(b), the ERR of $J = 0.05$ is smaller than that of $J = 0.10$, leading to improved MRAE performance for $J = 0.05$. As a result, the slow convergence phenomenon with larger J values that can occur with the same training

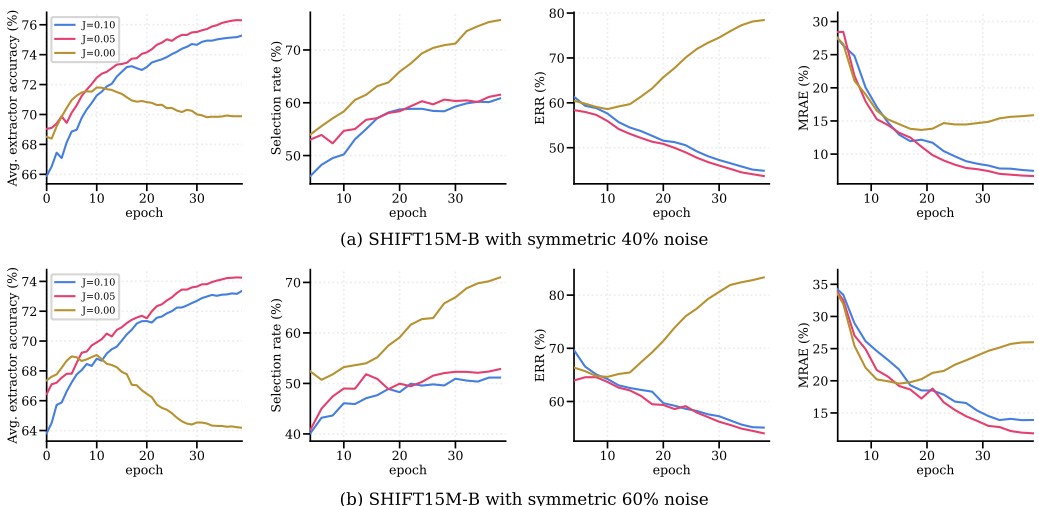

Figure 12: **Hyperparameter J analysis** compares the average accuracy of feature extractors, the Selection rate, ERR and MRAE on SHIFT15M-B with symmetric 40%, 60% noise.

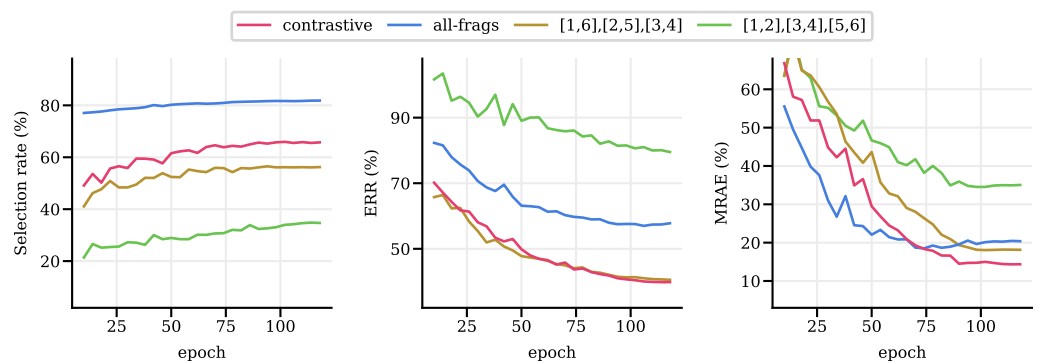

Figure 13: **Contrasting Fragment combination analysis** compares contrastive pairings ($[1, 4], [2, 5], [3, 6]$), all-fragments ($[1, 2, 3, 4, 5, 6]$), and other pairing methods ($[1, 2], [3, 4], [5, 6]$ and $[1, 6], [2, 5], [3, 4]$) on IMDB-Clean-B with 40% symmetric noise. All-fragments use a ResNet-34, while other pairing methods use ResNet-18 backbones.

time leads to relatively lower MRAE performance. Similar effects are observed in the SHIFT15M dataset, as depicted in Fig. 12(SHIFT15M-B).

## F.4 CONTRASTING FRAGMENTS COMBINATIONS

In Fig. 1(c), we offer deeper insights into our approach by comparing contrasting fragments ($[1, 4], [2, 5], [3, 6]$) against all-fragments ($[1, 2, 3, 4, 5, 6]$). In Fig. 13, we present the extended results with Selection rate, ERR, and MRAE alongside other pairing methods ($[1, 2], [3, 4], [5, 6]$ and $[1, 6], [2, 5], [3, 4]$).

The experiments involve training the feature extractors using either contrasting fragments, all-fragments, or alternative pairings. Notably, a single feature extractor is employed for all-fragments, whereas the paired grouping use a smaller feature extractor for each individual pair. Subsequently, sample selection is executed in accordance with the Mixture of Neighboring Fragments approach

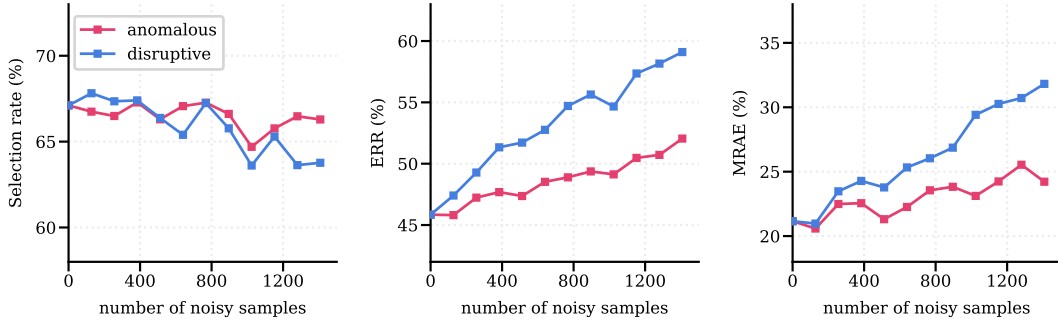

Figure 14: **Disruptive/anomalous noise analysis** displays the selection, ERR score and MRAE when disruptive or anomalous noisy samples are injected into the clean dataset. The experiments are based on IMDB-Clean-B.

(§ 2.3). When identifying the self-agreement $\alpha_f^{\text{self}}$ (Eq. 4), the contrasting pair $f^+$ is determined through contrastive pairing ($[1, 4], [2, 5], [3, 6]$).

In an optimal selection algorithm, the Selection rate should approach $100 -$ noise rate$(\%)$, with ERR and MRAE minimized. Across all evaluation metrics, the contrasting fragment pairing demonstrates superior performance compared to other methods. It is important to highlight that performance is poorest when the pairing is least distinguishable ($[1, 2], [3, 4], [5, 6]$) and moderate when the pairing is partially distinguishable ($[1, 6], [2, 5], [3, 4]$).

Furthermore, in Fig. 15, we utilize t-SNE to compare the feature extractors trained using contrasting pairs and all-fragments. The visual comparison clearly validates that representations trained with contrasting pairs exhibit significantly more distinguishable features.

### F.5 DISRUPTIVE VERSUS ANOMALOUS NOISE

To explore the impact of disruptive or anomalous samples, as depicted in Fig. 1(b) in the main manuscript, we conducted an analysis of Selection rate, Error Residual Rate (ERR), and Mean Relative Absolute Error (MRAE) performance while gradually introducing disruptive and anomalous noisy samples into the IMDB-Clean-B dataset.

Our study employ the IMDB-Clean-B dataset, comprising a fixed set of clean samples that represent 40% of the total dataset, alongside varying amounts of noisy samples. These noisy samples are classified into two distinct categories: disruptive and anomalous noise. The classification is determined by comparing their fragment ids, which incorporate the noisy label ($y$), with those containing the ground-truth label ($y^{\text{gt}}$).

To provide further clarification, let's consider an example: a sample with a ground-truth label has a fragment id of 1, but the noisy label has an assigned fragment id of 3, signifying a contrasting pair. In such cases, we designate it as a disruptive noise sample. Conversely, if a sample with a ground-truth label has an assigned fragment id of 1, and the noisy label's assigned fragment id is either 2 or 4, which does not form a contrasting pair, we classify it as an anomalous noise sample.

Fig. 14 demonstrates that disruptive noisy samples have a considerably more adverse impact on ERR and MRAE compared to anomalous noisy samples. Our contrastive fragmentation pair-based learning approach is advantageous in this regard, as it introduces anomalous noisy samples in lieu of many disruptive noisy samples, thereby facilitating learning with reduced interference.

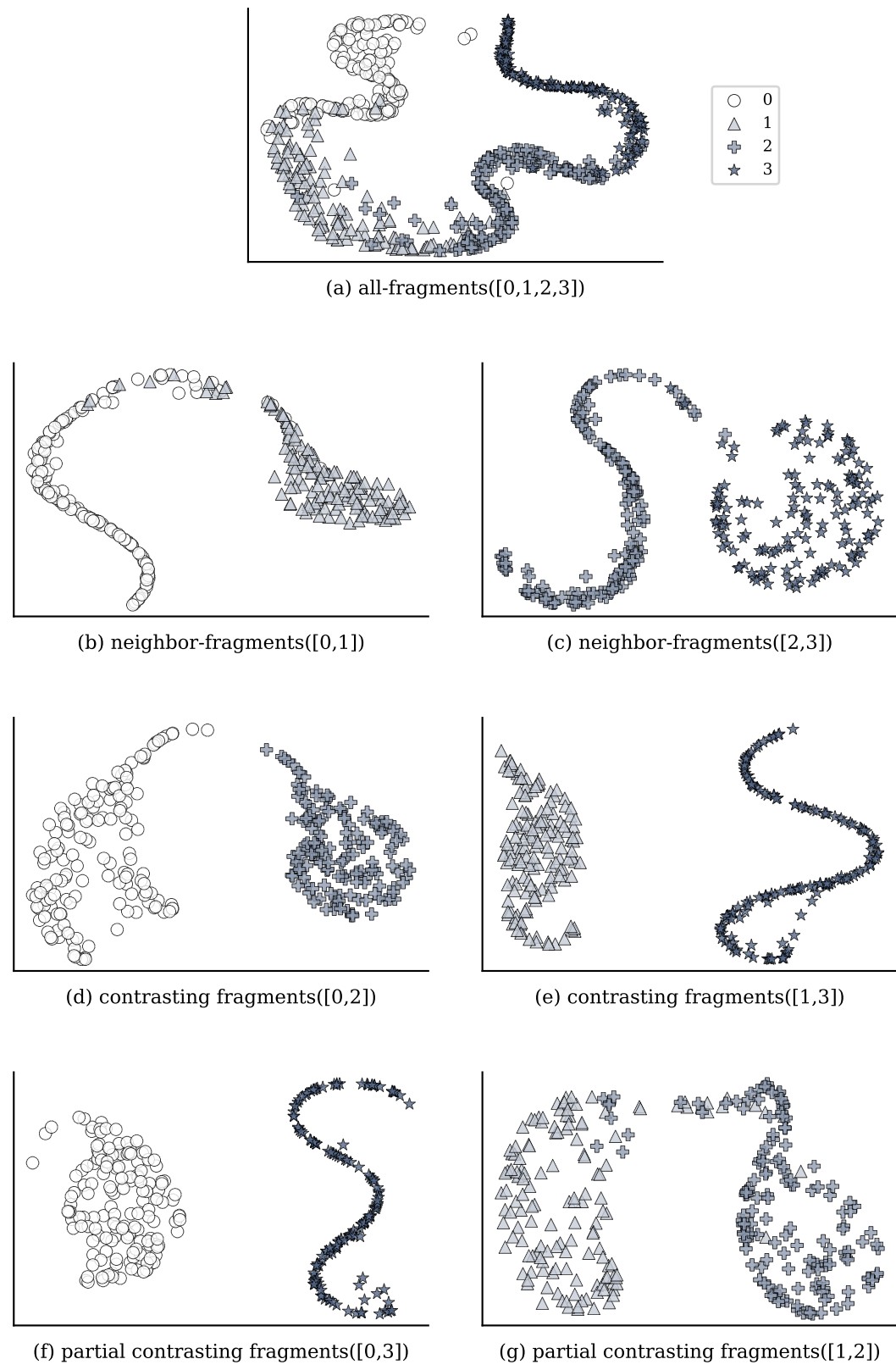

Figure 15: **Detailed Representation Depiction**. A detailed comparison of the contrasting fragment's effects via visualization of the penultimate feature of the discriminator using t-SNE. The experiments are based on IMDB-Clean-B.

| ablation and combinations | | | | | IMDB-Clean-B | | |
| --- | --- | --- | --- | --- | --- | --- | --- |
| | | | | | symmetric | Gaussian | |
| loss | backbone | jitter | C-Mixup | Co-teaching | 40 | 30 | 50 |
| MSE & MSE | ResNet-18 | | | | 22.37 | 21.28 | 45.15 |
| MSE & MSE | ResNet-18 | ✓ | | | 22.22 | 24.42 | 43.31 |
| CE & MSE | ResNet-18 | | | | 18.90 | 21.77 | 39.78 |
| CE & MSE | ResNet-18 | ✓ | | | 10.53 | 16.02 | 30.80 |
| CE & MSE | ResNet-34 | ✓ | | | 13.44 | 16.06 | 31.00 |
| CE & MSE | ResNet-18 | ✓ | ✓ | | 7.59 | 11.24 | 29.23 |
| CE & MSE | ResNet-18 | ✓ | | ✓ | 9.13 | 14.61 | 35.92 |
| SCE & MSE | ResNet-18 | | | | 18.37 | 20.80 | 38.10 |
| SCE & MSE | ResNet-18 | ✓ | | | 16.84 | 20.07 | 38.18 |
| SCE & MSE | ResNet-34 | ✓ | | | 14.97 | 18.95 | 36.12 |
| SCE & MSE | ResNet-18 | ✓ | ✓ | | 15.85 | 16.27 | 36.42 |
| SCE & MSE | ResNet-18 | ✓ | | ✓ | 13.19 | 18.32 | 41.02 |

Table 4: **Ablation and Combination Analysis.** The values are mean relative absolute error to the noise-free Vanilla model on the IMDB-Clean-B (Rothe et al., 2018) dataset, and lower values indicate better performances.

| ablation | | | IMDB-Clean-B | | |
| --- | --- | --- | --- | --- | --- |
| | | | symmetric | Gaussian | |
| $\alpha_f^{\text{self}}$ | $\alpha_f^{\text{ngb}}$ | $\mathcal{S}$ | 40 | 30 | 50 |
| ✓ | | $\mathcal{S}^p \cup \mathcal{S}^r$ | 13.66 | 15.90 | 32.95 |
| | ✓ | $\mathcal{S}^p \cup \mathcal{S}^r$ | 19.84 | 24.14 | 42.63 |
| ✓ | ✓ | $\mathcal{S}^p$ | 12.59 | 14.02 | 31.15 |
| ✓ | ✓ | $\mathcal{S}^r$ | 12.34 | 16.90 | 36.51 |
| ✓ | ✓ | $\mathcal{S}^p \cap \mathcal{S}^r$ | 11.87 | 14.76 | 34.03 |
| ✓ | ✓ | $\mathcal{S}^p \cup \mathcal{S}^r$ | 10.53 | 16.02 | 30.80 |

Table 5: **Ablation of Mixture of Neighboring Fragments.** The values are mean relative absolute error to the noise-free Vanilla model on the IMDB-Clean-B (Rothe et al., 2018) dataset, and lower values indicate better performances.

## F.6 SELECTION RATIO ANALYSIS BASED ON NOISE TYPES

In Fig. 16, we delineate the selected samples by FragSel on IMDB-Clean-B with symmetric 40% noise. Each selection ratio is calculated as,

$$\text{selection ratio} = \frac{\text{selected number}}{\text{total}}$$

where the 'selected number' and 'total' are one of clean/anomaly/disruptive, respectively. We can see that FragSel has a high clean sample selection ratio that improves with training, anomaly samples, and disruptive samples, which decay as training progresses. Resultantly, FragSel selects a higher-quality dataset as the training progresses.

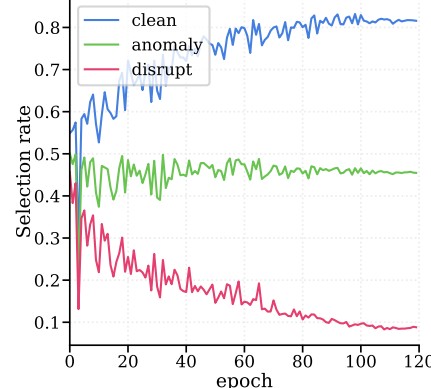

Figure 16: **Clean/Anomaly/Disruptive sample selection ratio**. The analysis is conducted on IMDB-Clean-B with symmetric 40% noise.

### F.7  ABLATION & COMBINATION ANALYSIS

In Table 4, we present a comprehensive study comparing the performance of Mean Squared Error (MSE), Cross-Entropy (CE), and Symmetric Cross Entropy (SCE) (Wang et al., 2019) losses in various ablation and combination experiments conducted on the IMDB-Clean-B dataset (Rothe et al., 2018), considering scenarios with 40% symmetric noise and two variations of Gaussian random noise, each having a maximum standard deviation of 30 and 50.

In the first ablation experiment, we utilize a regression feature extractor trained solely using the vanilla Mean Squared Error (MSE) loss. While this approach performs reasonably well in isolation, it falls slightly short in achieving discriminative feature extraction performance.

Subsequently, we illustrate the impact of jittering regularization through ablation on each of the losses. Notably, jittering regularization emerges as a crucial component for FragSel's performance, preventing the model from overfitting to the noisy labels.

Next ablation experiment entails replacing the ResNet-18 architecture with ResNet-34. The performance is enhanced when trained with SCE but decreases when trained with just CE. This suggests that FragSel could potentially benefit from a more powerful architecture, but it is not a necessity.

A significant advantage of FragSel lies in its compatibility with other approaches. We showcase its performance when combined with two additional techniques: C-Mixup (Yao et al., 2022) and Co-teaching (Han et al., 2018), which are also employed by CNLCU and Co-Selfie in our baseline. Co-teaching involves training the regression model while heuristically assuming that 25% of the original noise still exists in the data (*e.g.*, 40% original noise implies an assumption of 10% noise during Co-teaching regression). Additionally, we report the results of combining C-Mixup with our regression model. Empirical observations reveal that Co-teaching consistently provides significant benefits, while the impact of C-Mixup on performance varies depending on the scenario, but it overall performs best when used with CE.

Upon comparing CE and SCE for feature extractor training loss, we observe that CE, when combined with jitter regularization, synergizes better to exhibit much stronger performance compared to SCE.

In Table 5, we conduct an ablation analysis of the Mixture of Neighboring Fragments (§ 2.3). When evaluating neighborhood agreeability based solely on either the agreement of the current fragment ($\alpha_f^{\text{self}}$) or the neighboring fragment's agreement ($\alpha_f^{\text{ngb}}$), the ablation reveals that relying on the current fragment's agreement alone ($\alpha_f^{\text{self}}$) exhibited relatively stronger performance. Nevertheless, this approach still fell short of achieving a satisfactory level compared to considering both agreements, as defined in Eq. 6.

Next, as we consider sample selection based on both the predictive inference output and the representational inference output (referred to as the selected sample sets $\mathcal{S}^p$ and $\mathcal{S}^r$ respectively), we conduct an ablation study on these selected sample sets. This involves evaluating the results when determining the final selected sample set ($\mathcal{S}$) either individually, at the intersection, or at the union of $\mathcal{S}^p$ and $\mathcal{S}^r$. The findings indicate that utilizing samples solely from the predictive inference output ($\mathcal{S}^p$), or from the intersection of sample sets from both predictive and representational inference outputs ($\mathcal{S}^p \cap \mathcal{S}^r$), demonstrates notably strong performance, particularly at Gaussian 30% noise. However, overall, in line with FragSel, the union of sets ($\mathcal{S}^p \cup \mathcal{S}^r$) proves to be the most effective strategy.

### F.8  DISCRETIZED BASELINES

In Table 6, we present a discretized version of several strong baselines, including Sigua (Han et al., 2020), CNLCU (Xia et al., 2022), BMM (Arazo et al., 2019), Selfie/Co-Selfie (Song et al., 2019), MD-DYR-SH (Arazo et al., 2019), and CRUST (Mirzasoleiman et al., 2020).

The discretization process aligns with our fragmentation approach used for FragSel. We obtain selected samples at the end of every epoch to independently train the regression model. Additionally, we report performance with mixup (Zhang et al., 2018), a technique that proves beneficial for some baselines like Sigua (Han et al., 2020).

Notably, all baselines exhibit a deterioration in performance following discretization. However, Selfie/Co-Selfie (Song et al., 2019) stands out as the exception, showing an improvement in perfor-

|  | IMDB-Clean-B | | |
| :--- | :---: | :---: | :---: |
|  | symmetric | Gaussian | |
| noise rate (%) | 40 | 30 | 50 |
| CNLCU-S-D (Xia et al., 2022) | 55.71 | 64.71 | 79.59 |
| CNLCU-S-D + mixup (Xia et al., 2022) | 55.14 | 67.17 | 81.32 |
| CNLCU-H-D (Xia et al., 2022) | 37.76 | 51.36 | 76.40 |
| CNLCU-H-D + mixup (Xia et al., 2022) | 65.32 | 67.31 | 84.22 |
| Sigua-D (Han et al., 2020) | 56.17 | 61.67 | 66.08 |
| Sigua-D + mixup (Han et al., 2020) | 33.55 | 29.33 | 49.44 |
| BMM-D (Arazo et al., 2019) | 33.86 | 30.27 | 50.05 |
| MD-DYR-SH-D (Arazo et al., 2019) | 33.89 | 31.18 | 51.23 |
| CRUST-D (Mirzasoleiman et al., 2020) | 33.86 | 30.27 | 50.47 |
| CRUST-D + mixup (Mirzasoleiman et al., 2020) | 32.33 | 30.50 | 50.27 |
| Selfie-D (Song et al., 2019) | 31.50 | 24.86 | 47.46 |
| Selfie-D + mixup (Song et al., 2019) | 35.33 | 28.02 | 46.42 |
| Co-Selfie-D (Song et al., 2019) | 30.20 | 26.36 | 49.61 |
| Co-Selfie-D + mixup (Song et al., 2019) | 33.18 | 28.28 | 52.20 |
| FragSel-D (Ours) | 12.64 | 15.70 | 33.36 |
| Co-FragSel-D (Ours) | 9.45 | 14.87 | 35.88 |

Table 6: **Discretized Baseline Analysis.** Mean Relative Absolute Error to the noise-free Vanilla model of discretized versions of strongly performing models on the IMDB-Clean-B (Rothe et al., 2018) dataset. Lower is better.

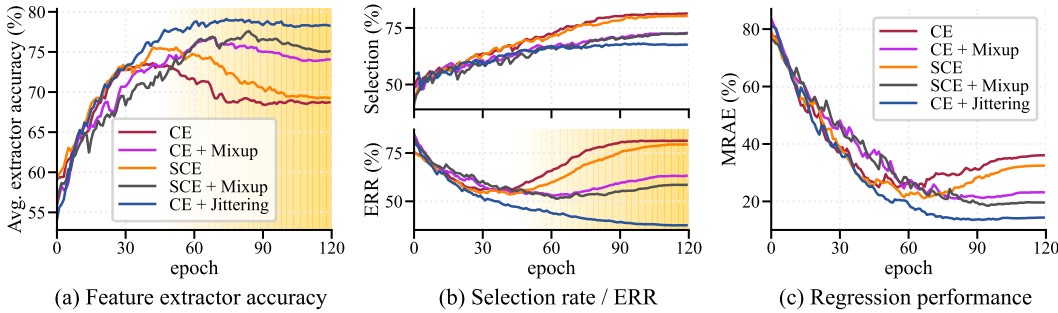

(a) Feature extractor accuracy     (b) Selection rate / ERR     (c) Regression performance

Figure 17: **Comparison of regularization methods**. Compared to other regularization methods, neighborhood jittering demonstrates superior performance in (a) feature extractor test accuracy, (b) ERR, and (c) performance in regression. The analysis is conducted on IMDB-Clean-B with symmetric 40% noise.

mance after discretization. Interestingly, Sigua is the sole method that benefits from mixup (Zhang et al., 2018) training.

### F.9 COMPARISON WITH NEIGHBORHOOD JITTERING AND OTHER REGULARIZATION METHODS

In Table 17, we compare neighborhood jittering with other regularization methods that can be applied to classification-based feature extractors (SCE with weight decay (Wang et al., 2019), mixup (Zhang et al., 2018), and their combinations). In conclusion, neighborhood jittering exhibits the strongest performance in feature extractor test accuracy, ERR, and MRAE, among other regularization methods. It is observed that ERR and MRAE improve in line with the performance of the feature extractor.

### F.10 EXTENDED SELECTION RATE/ERR/MRAE COMPARISON AND ANALYSIS

In addition to presenting the Selection rate, Error Residual Rate (ERR) and Mean Relative Absolute Error(MRAE) for symmetric 40%, Gaussian 30, and Gaussian 50 noise experiments on the IMDB-Clean-B dataset in the main manuscript, we have included results for all noise types, along with

additional baselines (CNLCU-H, Sigua, BMM, DY-S, AUX, Selfie, Coselfie), in both Fig. 18 and Fig. 19.

As mentioned in § 4.2, the ideal scenario for selection and refurbishment methods involves achieving a high selection rate while maintaining a low ERR, resulting in a reduced mean relative absolute error (MRAE). We examine the relationship between the selection rate, ERR, and MRAE based on Fig. 18(b). As training progresses, FragSel and other selection methods (CNLCU-H, Sigua, BMM, DY-S) approach the ideal condition, resulting in an improving trend in MRAE. FragSel, in particular, comes closest to the ideal scenario, resulting in superior MRAE performance.

The most unfavorable scenario arises when there is a low selection rate coupled with a high ERR. Selfie exemplifies the scenario in Fig. 18(b), which is connected to a relatively worse MRAE.

The scenarios of the low selection rates with low ERR and the high selection rates with high ERR can be further examined using CNLCU-H and BMM. CNLCU-H demonstrates superior selection quality in terms of ERR, while BMM exhibits a higher quantity in the selection rate. This quality/quantity trade-off is linked to the observation that CNLCU-H and BMM show similar MRAE performance in Fig. 18(b). Additionally, Fig. 19(a) reveals that the selection rate gap widens, while the ERR gap narrows when compared to Fig. 18(b). This is associated with BMM outperforming CNLCU-H in terms of the MRAE.

It's important to note that, rather than employing the selection rate and ERR as indicators for MRAE, as discussed above, these metrics offer valuable insights when assessing selected or refurbished samples directly independent of any potential regularizing effects introduced by the underlying regression model.

In addtion, upon a detailed analysis of the figures, it becomes evident that Co-FragSel consistently achieves the lowest ERR across a wide range of noise types. Notably, it maintains a Selection rate of above 40% even in the presence of severe noise conditions, which leads to outstanding MRAE performance.

## F.11 VARIANCE ANALYSIS

In Fig. 20, we plot the variance of three unique random seed experiments on all six noise types (symmetric 20%/40%/60%/80%, Gaussian 30/50) on the IMDB-Clean-B dataset. To declutter the graph, we compare it against the top two best-performing baselines under each noise type.

## F.12 STANDARD MEAN ABSOLUTE ERROR

In Tables 7, we report the standard mean absolute error within the respective label ranges for each dataset.

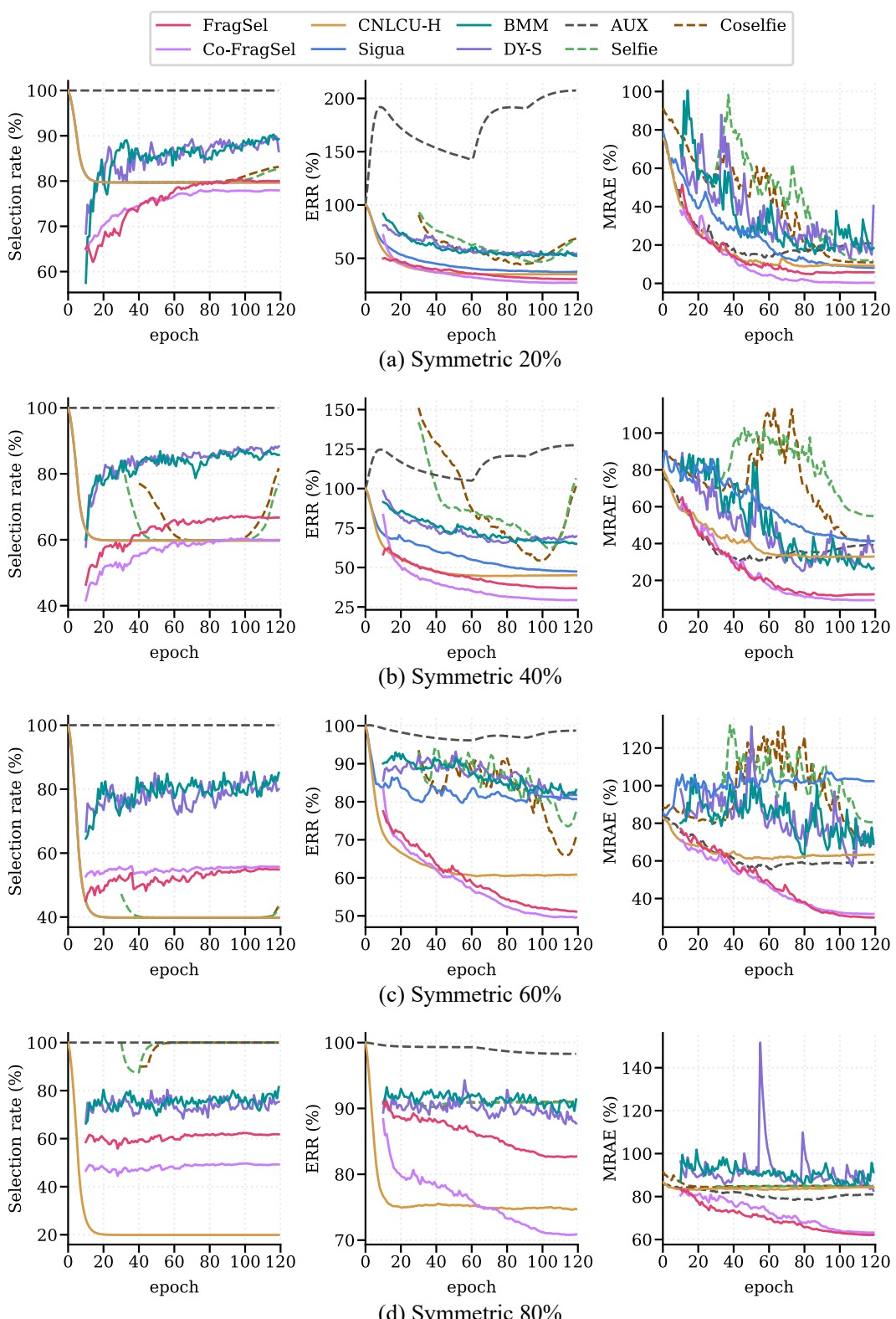

Figure 18: **Selection, ERR and MRAE comparison** of FragSel, Co-FragSel and filtering/refurbishment baselines on IMDB-Clean-B with symmetric 20%(a), 40%(b), 60%(c) and 80%(d) noise, repectively.

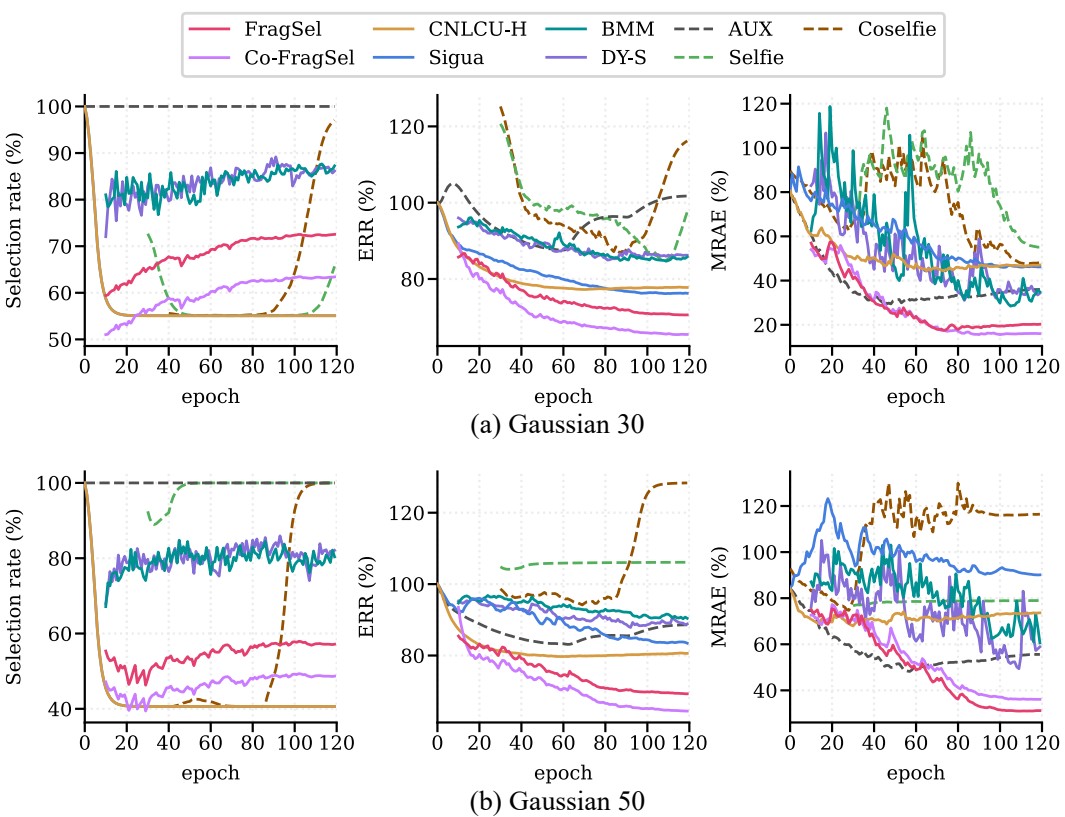

Figure 19: **Selection, ERR and MRAE comparison** of FragSel, Co-FragSel and filtering/refurbishment baselines on IMDB-Clean-B with Gaussian 30(a) and Gaussian 50(b) noise, repectively.

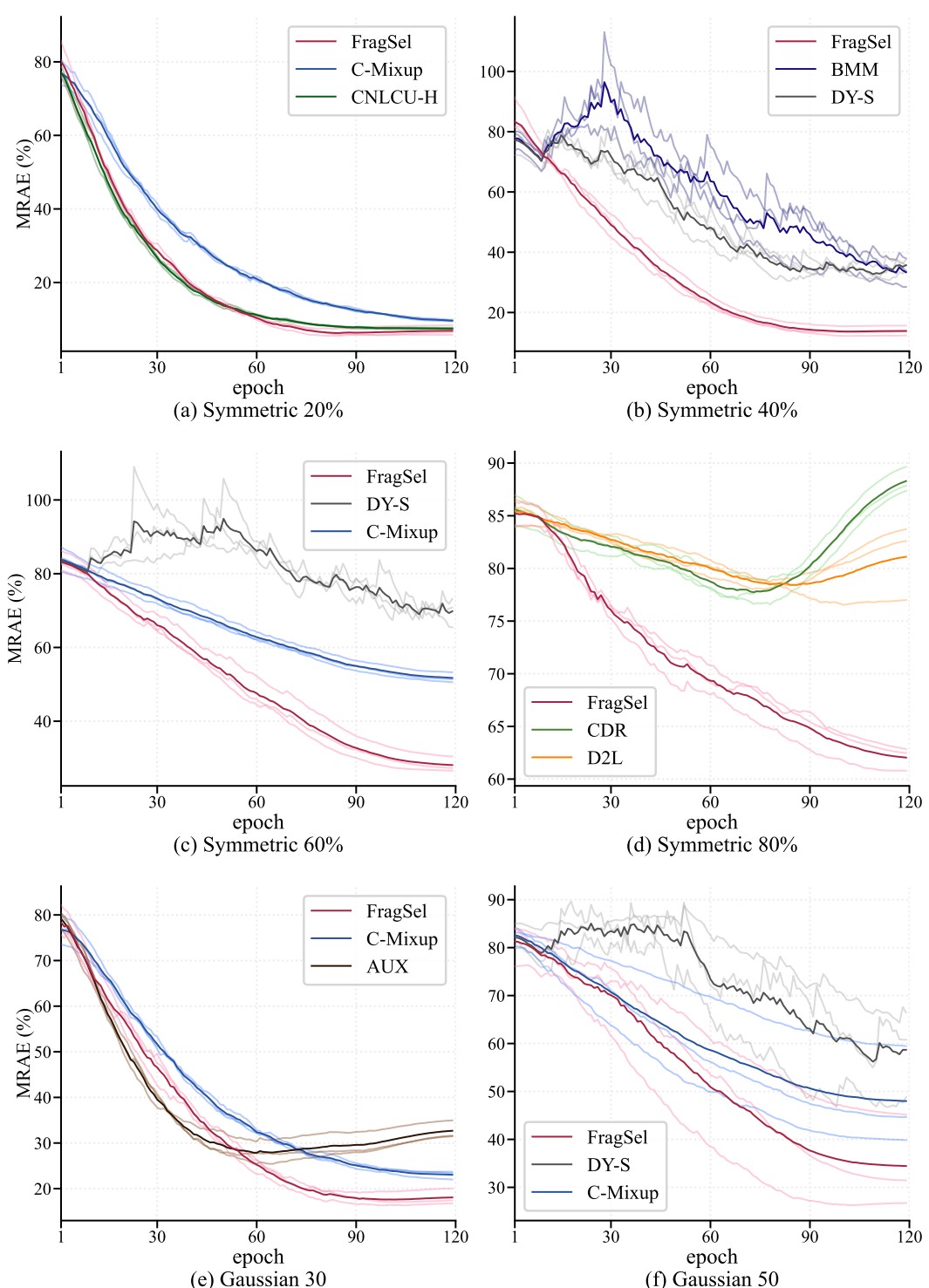

Figure 20: **Variance Analysis** of three unique random seed experiments on IMDB-Clean-B. The top two best-performing baselines under each noise type are reported.

Table 7: **Standard Mean Absolute Error** on the AFAD-B, IMDB-Clean-B, SHIFT15M-B, MSD-B dataset. Lower is better. The results are the mean of three random seed experiments. The best and the second best methods are respectively marked in red and blue. CNLCU-S/H, Co-Selfie, and Co-FragSel use dual networks to teach each other as done in Han et al. (2018). SPR (Wang et al., 2022) fails to run for SHIFT15M-B due to excessive memory consumption.

| | AFAD-B | | | | | | IMDB-Clean-B | | | | | |
|---|---|---|---|---|---|---|---|---|---|---|---|---|
| | symmetric | | | | Gaussian | | symmetric | | | | Gaussian | |
| noise rate | 20 | 40 | 60 | 80 | 30 | 50 | 20 | 40 | 60 | 80 | 30 | 50 |
| Vanilla | 4.75 | 5.22 | 5.68 | 6.22 | 5.59 | 6.04 | 8.11 | 9.22 | 10.70 | 12.32 | 8.86 | 10.50 |
| CNLCU-S | 4.82 | 5.23 | 5.75 | 6.17 | 5.67 | 6.11 | 10.57 | 11.64 | 12.77 | 12.97 | 12.81 | 12.72 |
| CNLCU-H | 4.55 | 5.05 | 5.91 | 6.29 | 5.89 | 6.24 | 7.46 | 9.16 | 11.39 | 12.76 | 10.24 | 11.54 |
| Sigua | 4.60 | 5.26 | 6.23 | 6.50 | 6.19 | 6.35 | 7.67 | 10.21 | 12.40 | 12.96 | 11.25 | 12.39 |
| SPR | 4.77 | 5.16 | 5.67 | 6.22 | 5.58 | 6.07 | 8.00 | 9.25 | 10.82 | 12.53 | 8.78 | 10.55 |
| BMM | 4.59 | 5.00 | 6.04 | 6.36 | 5.69 | 6.26 | 7.60 | 8.49 | 10.87 | 12.60 | 8.68 | 10.98 |
| DY-S | 4.64 | 5.02 | 5.74 | 6.33 | 5.40 | 6.23 | 7.71 | 8.51 | 10.47 | 12.44 | 8.71 | 10.10 |
| C-Mixup | 4.46 | 4.99 | 5.52 | 6.17 | 5.40 | 5.95 | 7.60 | 8.92 | 10.54 | 12.35 | 8.52 | 10.27 |
| RDI | 4.81 | 5.29 | 6.05 | 6.39 | 5.97 | 6.27 | 8.13 | 9.03 | 10.89 | 12.57 | 8.78 | 10.57 |
| CDR | 4.79 | 5.16 | 5.75 | 6.23 | 5.64 | 6.05 | 8.20 | 9.23 | 10.81 | 12.25 | 8.97 | 10.60 |
| D2L | 4.75 | 5.24 | 5.70 | 6.28 | 5.60 | 6.09 | 8.17 | 9.35 | 10.86 | 12.31 | 9.03 | 10.65 |
| AUX | 4.61 | 5.17 | 5.70 | 6.20 | 5.57 | 6.04 | 7.86 | 9.00 | 10.64 | 12.35 | 8.61 | 10.44 |
| Selfie | 5.08 | 5.43 | 6.26 | 6.42 | 6.34 | 6.55 | 8.90 | 10.74 | 12.53 | 12.85 | 11.22 | 12.43 |
| Co-Selfie | 4.98 | 5.34 | 6.07 | 6.42 | 6.13 | 6.65 | 8.63 | 10.48 | 11.69 | 12.87 | 10.65 | 12.20 |
| Superloss | 4.66 | 5.14 | 5.64 | 6.27 | 5.54 | 6.21 | 8.62 | 10.16 | 11.67 | 12.63 | 10.75 | 11.41 |
| **FragSel-R** | 4.56 | 4.95 | 5.55 | 5.96 | 5.30 | 5.82 | 7.59 | 8.57 | 10.08 | 11.74 | 8.50 | 10.26 |
| **Co-FragSel-R** | 4.44 | 4.79 | 5.32 | 5.97 | 5.29 | 5.81 | 7.17 | 8.11 | 9.79 | 11.73 | 8.28 | 10.39 |
| **FragSel-D** | 4.46 | 4.70 | 5.04 | 5.84 | 5.10 | 5.53 | 7.34 | 7.87 | 8.89 | 11.26 | 8.08 | 9.31 |
| **Co-FragSel-D** | 4.37 | 4.66 | 5.07 | 5.82 | 5.10 | 5.57 | 7.09 | 7.64 | 8.97 | 11.27 | 8.02 | 9.49 |

| | SHIFT15M-B | | | | | | MSD-B | | | | | |
|---|---|---|---|---|---|---|---|---|---|---|---|---|
| | symmetric | | | | Gaussian | | symmetric | | | | Gaussian | |
| noise rate | 20 | 40 | 60 | 80 | 30 | 50 | 20 | 40 | 60 | 80 | 30 | 50 |
| Vanilla | 7.47 | 8.08 | 8.70 | 9.34 | 7.30 | 7.89 | .5918 | .6475 | .7199 | .7974 | .5848 | .6328 |
| CNLCU-S | 7.74 | 8.18 | 8.51 | 9.21 | 7.90 | 8.28 | .5475 | .5798 | .6644 | .7983 | .5727 | .6151 |
| CNLCU-H | 7.28 | 7.73 | 8.22 | 9.32 | 7.46 | 7.92 | .5483 | .5740 | .6032 | .7098 | .5747 | .5972 |
| Sigua | 7.32 | 7.81 | 8.64 | 9.39 | 7.56 | 8.04 | .5538 | .5861 | .6416 | .8248 | .5839 | .6145 |
| SPR | - | - | - | - | - | - | .5854 | .6462 | .7293 | .7961 | .5741 | .6308 |
| BMM | 7.33 | 7.70 | 8.13 | 8.68 | 7.37 | 7.75 | .5649 | .6031 | .6747 | .7849 | .5757 | .6116 |
| DY-S | 7.34 | 7.67 | 8.14 | 8.84 | 7.32 | 7.77 | .5653 | .5908 | .6487 | .7394 | .5728 | .6005 |
| C-Mixup | 7.50 | 7.95 | 8.50 | 9.19 | 7.25 | 7.84 | .5673 | .6185 | .6929 | .7704 | .5630 | .6067 |
| RDI | 7.53 | 8.08 | 8.67 | 9.33 | 7.33 | 7.89 | .6618 | .7113 | .7588 | .8174 | .6517 | .6992 |
| CDR | 7.50 | 8.07 | 8.70 | 9.31 | 7.34 | 7.89 | .5896 | .6444 | .7262 | .7978 | .5836 | .6393 |
| D2L | 7.48 | 8.08 | 8.67 | 9.33 | 7.28 | 7.92 | .5857 | .6559 | .7243 | .8018 | .5769 | .6317 |
| AUX | 7.38 | 8.01 | 8.67 | 9.35 | 7.19 | 7.83 | .5802 | .6462 | .7167 | .7966 | .5753 | .6312 |
| Selfie | 7.18 | 7.55 | 8.37 | 9.46 | 7.23 | 7.64 | .5546 | .5927 | .6574 | .7976 | .6253 | .6787 |
| Co-Selfie | 7.64 | 7.97 | 9.05 | 9.54 | 7.77 | 8.38 | .5447 | .5709 | .5923 | .7407 | .5839 | .6187 |
| Superloss | 7.22 | 7.69 | 8.44 | 9.26 | 7.23 | 7.76 | .5460 | .6052 | .6733 | .7959 | .5706 | .6362 |
| **FragSel-R** | 7.13 | 7.51 | 7.96 | 8.61 | 7.19 | 7.60 | .5510 | .5778 | .6212 | .7110 | .5620 | .5843 |
| **Co-FragSel-R** | 6.97 | 7.37 | 7.81 | 8.50 | 7.12 | 7.51 | .5451 | .5654 | .6031 | .6902 | .5587 | .5843 |
| **FragSel-D** | 7.02 | 7.27 | 7.58 | 8.15 | 7.10 | 7.40 | .5499 | .5738 | .6081 | .6747 | .5598 | .5822 |
| **Co-FragSel-D** | 6.91 | 7.23 | 7.59 | 8.14 | 7.06 | 7.44 | .5432 | .5631 | .5941 | .6590 | .5562 | .5796 |

---

**Algorithm 1** Fragmented Selection

---

**Input:** Train data $\mathcal{D} = \{\mathcal{X}, Y\}$, Fragment number $F$, KNN parameter $K$, Jitter $J$, Total epochs $N$

$\mathcal{S}, \mathcal{S}^p, \mathcal{S}^r = \{\}, \{\}, \{\}$      # selected samples
$\Theta = \{\theta_{0,0} \ldots \theta_{i,j}\}$      # feature extractors

$\mathcal{D}_{1 \ldots F} = Fragmentation(\mathcal{D})$      # § 2.1. 1
$\mathcal{P} = ContrastivePairing(\mathcal{D}_{1 \ldots F})$      # § 2.1. 2∼4
**for** $n$ **to** $N$ **do**
   # train feature extractors
   $\mathcal{P}^{jitter} = NeighborhoodJittering(\mathcal{D}, F, J)$      # neighborhood jittering (§ 2.4)
   **for** $(\mathcal{D}_i^{jitter}, \mathcal{D}_j^{jitter})$ **in** $\mathcal{P}^{jitter}$ **do**
     $\mathcal{D}_{i,j}^{jitter} = \mathcal{D}_i^{jitter} \cup \mathcal{D}_j^{jitter}$
     train $p(f; \theta_{i,j}, \mathcal{D}_{i,j}^{jitter})$
   **end for**

   # obtain $\mathcal{S}^p, \mathcal{S}^r$
   **for** $(x, y)$ **in** $\mathcal{D}$ **do**
     **for** $f = 1$ **to** $F$ **do**
       calculate $\eta_f(y)$      # neighborhood prior (Eq. 2)
       calculate $\alpha_f^p(x; \mathcal{D}_{1 \ldots F}, \Theta)$      # pred. neighborhood agreeability (Eq. 6)
       calculate $\alpha_f^r(x; \mathcal{D}_{1 \ldots F}, \Theta)$      # repr. neighborhood agreeability (Eq. 6)
     **end for**
     $p^p(s|x, y, \mathcal{D}_{1 \ldots F}; \Theta) = \sum_f^F \eta_f(y) \alpha_f^p(x; \mathcal{D}_{1 \ldots F}, \Theta)$ # pred. sample % (Eq. 1)
     $p^r(s|x, y, \mathcal{D}_{1 \ldots F}; \Theta) = \sum_f^F \eta_f(y) \alpha_f^r(x; \mathcal{D}_{1 \ldots F}, \Theta)$ # repr. sample % (Eq. 1)
     sample $\{u^p, u^r\} \sim uniform(0, 1)$
     **if** $p^p(s|x, y, \mathcal{D}_{1 \ldots F}; \Theta) > u^p$ **then**
       $\mathcal{S}^p = \mathcal{S}^p \cup (x, y)$
     **end if**
     **if** $p^r(s|x, y, \mathcal{D}_{1 \ldots F}; \Theta) > u^r$ **then**
       $\mathcal{S}^r = \mathcal{S}^r \cup (x, y)$
     **end if**
   **end for**

   # union filtered samples $(\mathcal{S}^p, \mathcal{S}^r)$
   $\mathcal{S} = \mathcal{S}^p \cup \mathcal{S}^r$

**end for**

---

