# OpenReview forum: "FragSel: Fragmented Selection for Noisy Label Regression"
_ICLR.cc/2024/Conference — Submitted to ICLR 2024_

### Official Review · Reviewer_KgTt · 2023-10-24

**Soundness:** 3 good
**Presentation:** 3 good
**Contribution:** 3 good
**Rating:** 6
**Confidence:** 5

**Summary:**

This paper studied regression learning with noisy labels, which is a seldom explored but important problem for machine learning. To address the problem, this paper proposed a novel noise-robust method by performing sample selection via a characteristic that data points similar in the feature space are likely to have similar labels. In addition, a neighborhood jittering regularization is used to improve the robustness. Experimental results confirmed the superiority of the proposed method.

**Strengths:**

1. The studied problem is highly valuable in real-world applications while seldom explored. This work used noisy regression benchmarks from various domains, which fully demonstrated the application potential of the proposed method.
2. The proposed method is a reasonable solution that makes use of the orderly relationships within the label and feature spaces.
3. The discussions and ablation analyses are thorough, making the effectiveness of the proposed method convincing.

**Weaknesses:**

1. Some important baselines are missing. For example, [1] is a nice baseline for regression learning with noisy labels. [2] performed bounding box correction by minimizing the discrepancy between two classifiers. Besides, I think there are some other works in noise-robust object detection that consider regression learning with noisy labels.
2. Some highly related references in noisy label learning are missing. For example, the transition matrix methods [3-5], and the hybrid methods [6,7].
3. The description of the proposed algorithm procedure and the experiment setting can be introduced more clearly. I have some questions and suggestions: 1) Are the prediction-based or representation-based sample selections used together in the proposed method? If not, when to use the prediction-based sample selection, and when to representation-based one? 2)  How to inject symmetric label noise in regression labels? 3) The pseudo-code of the proposed algorithm will help a lot for the readers who want to understand the detailed design.


[1] Superloss: A generic loss for robust curriculum learning. NeurIPS 2020

[2] Towards noise-resistant object detection with noisy annotations. arXiv 2020

[3] Dual T: Reducing Estimation Error for Transition Matrix in Label-noise Learning. NeurIPS 2020

[4] Part-dependent Label Noise: Towards Instance-dependent Label Noise. NeurIPS 2020

[5] Estimating Noise Transition Matrix with Label Correlations for Noisy Multi-Label Learning. NeurIPS 2022

[6] Selective-Supervised Contrastive Learning with Noisy Labels. CVPR 2022

[7] Ngc: A unified framework for learning with open-world noisy data. ICCV 2021

**Questions:**

I think this work is a nice work if the authors can address my concerns above.

---

> ### Author Response · Authors · 2023-11-20
> **Official Comment by Submission3113 Authors**
>
> **W1. Additional baselines [1]? Consider object detection-based baselines?**
>
> In accordance with the reviewer's recommendation, we incorporate [1] into our main table. However, since [1] lacks an official code release, we developed it from scratch, making approximations regarding the LambertW function. We also initiated contact with the authors and will update the table with the official implementation if necessary.
>
> While examining the baseline methods, we also explored the application of the aforementioned object detection techniques to data with noisy annotations. However, our analysis revealed that these methods are technically not well-suited for the broader regression task. This is primarily due to the need for a classifier specifically trained for object category classification or the requirement of bounding box proposals from a region proposal network.
>
> Specifically, researchers such as [3, 4, 6] utilize region proposal networks to generate bounding box proposals. They leverage these proposals to selectively choose clean labels or re-weight the training samples. However, because this approach necessitates an auxiliary model in the proposal generation process, it cannot be directly applied in the context of regression tasks.
>
> On the other hand, [2-5] employs the object detector's classifier to update or assess the quality of bounding boxes. By evaluating the confidence or consistency of the bounding box through the classification output, this approach helps mitigate the impact of noisy labels. However, implementing a similar approach in the context of regression tasks would require the inclusion of an auxiliary co-trained task.
>
> [1] Castells, T., et al. Superloss: A generic loss for robust curriculum learning. In NeurIPS, 2020.
>
> [2] Li, J., et al. Towards noise-resistant object detection with noisy annotations. arXiv preprint arXiv:2003.01285, 2020.
>
> [3] Liu, C., et al. Robust object detection with inaccurate bounding boxes. In ECCV, 2022.
>
> [4] Schubert, M., et al. Identifying label errors in object detection datasets by loss inspection. arXiv preprint arXiv:2303.06999, 2023.
>
> [5] Gao, J., et al. Note-rcnn: Noise tolerant ensemble rcnn for semi-supervised object detection. In ICCV, 2019.
>
> [6] Mao, J., et al. Noisy annotation refinement for object detection. British Machine Vision Conference, 2021.
>
> **W2. Include Transition matrix and Hybrid methods in related works.**
>
> We appreciate the valuable insight of the related works! We appropriately cited them within the main body of the manuscript and dedicated additional sections for Transition Matrix, Object detection, and Hybrid Methods in the Appendix.
>
> **W3Q1. Clarify the combining of prediction and representation-based selections.**
>
> We revise our notation to follow our Ablation study in Appendix E.6 (Table 5) by introducing $S^p$ and $S^r$, where $p$ represents prediction and $r$ represents regression. In FragSel, we utilize the union of $S^p$ and $S^r$, denoted as $S^p \cup S^r$.
>
> **W3Q2. How to inject symmetric label noise in regression labels?**
>
> We adhere to the standard classification setting’s symmetric label noise [1], where a percentage of samples are uniformly flipped into other labels.
>
> [1] Yi, K., & Wu, J. Probabilistic end-to-end noise correction for learning with noisy labels. In CVPR, 2019.
>
> **W3Q3. Pseudo-code of FragSel**
>
> As per the reviewer’s suggestion, we include a pseudo-code of the proposed algorithm in the Appendix.

---

### Official Review · Reviewer_yzhv · 2023-10-31

**Soundness:** 3 good
**Presentation:** 3 good
**Contribution:** 3 good
**Rating:** 6
**Confidence:** 2

**Summary:**

The paper presents FragSel, a method for improving regression methods in the presence of noisy labels. This method uses a simple technique to learn better representations by training over maximally distance subsets, and the authors shown strong performance on an array of standard benchmarks for label noise.

**Strengths:**

- The overall communication is clear and straightforward, and the framing of the paper is evident and easy to understand throughout.
- The presented method FragSel is novel yet relatively simple, leading to an effective method for improving regression in the context of label noise that is straightforward to reproduce.
- The evaluation section is quite thorough, using a wide variety of benchmarks, noise methods, and evaluation metrics to assess the quality of their method.

**Weaknesses:**

- No real concerns are present, though I am not particularly well-versed in the literature on this topic so it is hard for me to assess if this work is sufficiently different from previous works.

**Questions:**

N/A

---

> ### Author Response · Authors · 2023-11-20
> **Official Comment by Submission3113 Authors**
>
> **W1. No real concerns are present, though I am not particularly well-versed in the literature on this topic so it is hard for me to assess if this work is sufficiently different from previous works.**
>
> We would like to provide a concise overview of the most relevant prior research in the field of noisy label learning that pertains to our core algorithm, FragSel. Specifically, we will focus on studies that incorporate neighborhood considerations and mixture models.
>
> As mentioned in the reviewer PSFZ's response to W2, the term 'Neighborhood' holds a prominent position within the extensive body of literature concerning noisy labels [1-6]. This prominence is substantiated by its proven efficacy in the selection of confident samples from uncalibrated neural network outputs. Nevertheless, its utilization in the context of noisy labeled regression remains an area that has yet to be investigated.
>
> Furthermore, our study presents several significant deviations from prior research. Of particular importance is that every distinctive characteristic of FragSel is deeply rooted in its fundamental dependence on 'contrastive fragments'. This dependence, in turn, gives rise to independent contrastive training as well as a mixture-based probabilistic selection framework.
>
> To elaborate further:
> - The mixture-based probabilistic framework for contrasting neighboring representations is distinctive because it incorporates a prior weighting based on relative distances among the mixtures (fragments) and includes a two-part agreement (self & neighboring) within each individual mixture and among them.
> - Our approach also involves contrastive fragment-based training of the representations, resulting in improved neighbor representations and, consequently, enhanced sampling techniques.
>
> Recently, [7] proposed a method to leverage noisy labels for anomaly detection. It uses a Mixture of Experts (MoE) to capture the similarities among noisy labels by sharing most model parameters while encouraging specialization by building expert sub-networks in the final MoE layer before the output layer.
>
> On the contrary, our model does not only account for anomalies, but all noise in general, using experts which do not share any parameters, but rather employ the independently trained experts for an ensemble effect for better robust filtering. Also, we uniquely use contrastive fragmentation to group the fragments for better learning of distinguishable representations and employ a mixture model on the fragment groups to collectively filter clean samples based on neighborhood agreements.
>
> [1] Li, J., et al. Neighborhood collective estimation for noisy label identification and correction. In ECCV, 2022.
>
> [2] Zhu, Z., et al. Detecting corrupted labels without training a model to predict. In ICML, 2022.
>
> [3] Shao, H. C., et al. Ensemble learning with manifold-based data splitting for noisy label correction. IEEE Transactions on Multimedia, 2022.
>
> [4] Wu, P., et al. A topological filter for learning with label noise. In NeurIPS, 2020.
>
> [5] Iscen, A., et al. Learning with neighbor consistency for noisy labels. In CVPR, 2022.
>
> [6] Xu, R., et al. Neighborhood-regularized self-training for learning with few labels. In AAAI, 2023.
>
> [7] Zhao, Y., et al. Admoe: Anomaly detection with mixture-of-experts from noisy labels. In AAAI, 2023.

---

### Official Review · Reviewer_PSFZ · 2023-11-01

**Soundness:** 3 good
**Presentation:** 3 good
**Contribution:** 2 fair
**Rating:** 5
**Confidence:** 4

**Summary:**

This papaer solves the problem of noisy label for regression task. It focuses on sample selection methodologies. It solves noisy label regression more well by (1) pairing samples with contrastive features, (2) considering neighbor agreement, and (3) neighborhood jittering. Additionally, this paper suggests new benchmark dataset for noisy label regression.

**Strengths:**

- Curate a new benchmark dataset for regression task, and evaluate current benchmarks.
- Apply graph structure to find the constasting pairs of dataset.
- Suggest a new metric called Error Residual Ratio (ERR).

**Weaknesses:**

- Figure 1 is hard to understand. It includes too much information that has not yet been explained.
- I think it is already quite well known that samples with similar features tend to exhibit similar labels, and many studies have assumes that properties; the validity of Semi supervised learning, pseudo labeling stems from this assumption. Therefore, I think that the novelty of this paper may be limited from several previous sample selection based methods, since I think the method proposed in this paper is the combinations of the previous studies (suggested in the classification task).
- I cannot understand yet why the method the authors suggests fits especially for the regression task. Can't it be applied to classification task?

**Questions:**

- It is known that data points with similar features tend to exhibit similar label values. However, including noisy labeled data samples, it corrupts these similarity the model learns because the model tries to fit all data samples, which is also the problem of learning noisy data. Therefore, for managing noisy data, can we use the similar feature-similar label property as it is? Or should we use additional tricks for managing the problem?
- Why should we select samples with framentation? (empirically, okay. Any theoretical idea?)
- Number of fragmentation would matter...
- Selecting the clean subset of the data, I think some bias can be included (e.g. maybe samples whose features are severely biased to one class will be easily sampled, and if samples are located between two fragments, it may not be selected although it is clean.). Can we mitigate it?

---

> ### Author Response · Authors · 2023-11-20
> **Official Comment by Submission3113 Authors**
>
> **W1. Figure 1 Update**
>
> We thank the reviewer for the insight! We have relocated Figure 1 and incorporated supplementary information to improve reader comprehension.
>
> **W2. Label Feature Correlation is widely assumed, is FragSel a combination of previous studies from noisy label classification?**
>
> We concur that the assumption regarding the correlation between labels and features has been previously investigated in various domains, which is the reason why we dealt with a detailed section as an extended related section in the Appendix (C.1)! However, given the extensive prior research on noisy labeled learning within the context of classification, we wish to emphasize its significance as the primary characteristic to be researched when developing a solution for handling noisy labels in regression tasks.
>
> The terms 'Neighborhood' and 'Contrasting' are prevalent keywords within the extensive body of literature on noisy label classification [1-11], and their prominence is well-justified, given their effectiveness in selecting confident samples from uncalibrated neural network outputs. However, their application in the context of noisy labeled regression remains unexplored. In addition, we present several notable distinctions from previous research. Most importantly, every unique aspect of FragSel stems from its foundational reliance on **contrastive fragments**. It is what leads to the independent contrastive training as well as the mixture-based probabilistic selection framework.
>
> - Specifically, the mixture-based probabilistic framework of neighboring contrastive representations uniquely consists of a prior weighting based on relative distances amongst the mixtures (fragments), as well as a two-part agreement (self & neighboring) within the single mixture as well as amongst them.
>
> - The contrastive fragment-based training of the representations results in improved neighbor representations and, consequently, enhanced sampling
>
> [1] Li, J., et al. Neighborhood collective estimation for noisy label identification and correction. ECCV, 2022.
>
> [2] Zhu, Z., et al. Detecting corrupted labels without training a model to predict. ICML, 2022.
>
> [3] Shao, H. C., et al. Ensemble learning with manifold-based data splitting for noisy label correction. IEEE Transactions on Multimedia, 2022.
>
> [4] Wu, P., et al. A topological filter for learning with label noise. NeurIPS, 2020.
>
> [5] Iscen, A., et al. Learning with neighbor consistency for noisy labels. CVPR, 2022.
>
> [6] Xu, R., et al. Neighborhood-regularized self-training for learning with few labels. AAAI, 2023.
>
> [7] Li, S., et al. Selective-supervised contrastive learning with noisy labels. CVPR, 2022.
>
> [8] Li, J., et al. Learning from noisy data with robust representation learning. ICCV, 2021.
>
> [9] Ortego, D., et al. Multi-objective interpolation training for robustness to label noise. CVPR, 2021.
>
> [10] Zhang, X., et al. Codim: Learning with noisy labels via contrastive semi-supervsied learning. arXiv preprint arXiv:2111.11652, 2021.
>
> [11] Huang, Z., et al. Twin contrastive learning with noisy labels. CVPR, 2023.
>
>
> **W3. Why use FragSel only on Regression and not Classification?**
>
> We focus on the task of noisy label regression because it is a recurring challenge in commercial applications and possesses unique characteristics compared to classification. However, while extending this approach to classification is feasible, it may necessitate specific approximations and the incorporation of prior knowledge to implement contrastive fragmentation based on the label-feature relationship assumption. One avenue worth investigating involves leveraging the CLIP model [1] to acquire label embeddings for the purpose of quantifying the distance relationship within the label space.
>
> [1] Radford, A., et al. Learning transferable visual models from natural language supervision. In ICML, 2021.
>
> **Q1. Does label feature correlation still hold when learning with noisy data? Should we use additional tricks?**
>
> We also acknowledge that the similarity relations learned by the model can be susceptible to disruptions. Therefore, the adoption of a robust algorithm holds paramount significance [1, 2]!
>
> We gently remind that FragSel effectively addresses this challenge through a multifaceted approach. This encompasses data fragmentation to facilitate stable training via cross-entropy loss, the utilization of contrastive pairing, the incorporation of Mixture of Neighbor agreements, and the introduction of jittering techniques to enhance regularization.
>
> Furthermore, the incorporation of "additional tricks" can prove advantageous. Notably, FragSel exhibits compatibility with various existing techniques, as demonstrated in Table 4, including SCE, co-teaching, and cmixup.
>
> [1] Zhang, C., et al. Understanding deep learning requires rethinking generalization. In ICLR, 2017.
>
> [2] Li, J., et al. How does a neural network’s architecture impact its robustness to noisy labels?. In NeurIPS, 2021.

---

> ### Author Response · Authors · 2023-11-20
> **Official Comment by Submission3113 Authors**
>
> **Q2. Theory of Fragmentation**
>
> FragSel operates by partitioning data samples into fragments and leveraging trained feature extractors for sample selection through collective modeling. This approach can be interpreted as a theoretically well-grounded Mixture-of-Experts (MoE), in which individual experts focus on specific subspaces of the problem through data partitioning [1, 2]. Through fragmentation, we can take MoE’s advantages in terms of computational scalability, handling mixed-type data, and reducing output variance [1]. It is worth highlighting that, as each network is trained on a distinct training set, MoE can effectively circumvent concurrent failures, thereby preventing error propagation among networks and ultimately enhancing generalization performance [3].
>
> In addition to the theoretical benefits associated with MoE, FragSel also addresses the issue of feature extractors memorizing incorrect labels when dealing with the noisy label problem. This is achieved by incorporating both self and neighbor agreements (Eq. 5), which bolsters robustness by preventing coincidental failures among feature extractors.
>
> Furthermore, the use of contrastive pairing for fragments enhances the training of feature extractors. Maximizing the distances between paired fragments ensures a substantial margin between their representations, which is pivotal for guiding robust training and promoting model generalizability [4, 5, 6]. While FragSel capitalizes on the distinctiveness of contrastiveness between labels, it's important to note that there are other well-grounded approaches that leverage label correlations, such as pairing labels in multi-label classification [7, 8], constructing label relation graphs [9], or employing label embedding techniques [10].
>
> Moreover, the process of fragmentation introduces the possibility of transforming a regression problem into a classification problem. As theoretically analyzed in [11], while Mean Squared Error (MSE) loss overlooks the marginal entropy of representation, Cross-Entropy (CE) loss maximizes it. Consequently, classification offers a more stable training paradigm compared to regression. For a comprehensive understanding of these concepts, please refer to the response to Q3 from reviewer 3QwR for detailed explanations.
>
> [1] Yuksel, S. E., et al. Twenty years of mixture of experts. IEEE transactions on neural networks and learning systems, 2012.
>
> [2] Masoudnia, S., & Ebrahimpour, R. Mixture of experts: a literature survey. Artificial Intelligence Review, 2014.
>
> [3] Sharkey, A. J., & Sharkey, N. E. Combining diverse neural nets. The Knowledge Engineering Review, 1997.
>
> [4] Shawe-Taylor, J., & Cristianini, N. Robust bounds on generalization from the margin distribution. 1998.
>
> [5] Grønlund, A., et al. Margin-based generalization lower bounds for boosted classifiers. In NeurIPS, 2019.
>
> [6] Grønlund, A., et al. Near-tight margin-based generalization bounds for support vector machines. In ICML, 2020.
>
> [7] Ghamrawi, N., & McCallum, A. Collective multi-label classification. ACM international conference on Information and knowledge management. 2005.
>
> [8] Read, J., et al. Classifier chains for multi-label classification. Machine learning, 2011.
>
> [9] Deng, J., et al. Large-scale object classification using label relation graphs. In ECCV, 2014.
>
> [10] Akata, Z., et al. Label-embedding for attribute-based classification. In CVPR, 2013.
>
> [11] Boudiaf, M., et al. A unifying mutual information view of metric learning: cross-entropy vs. pairwise losses. In ECCV, 2020.
>
> **Q3. Number of Fragments**
>
> We concur that determining the optimal number of fragments can be an important consideration. This is why we have included a discussion of the current limitations of FragSel in the Appendix, as well as an exploration of feasible approaches to mitigate these limitations. We reiterate that determining the optimal number of fragments without empirical search is a challenging task with related topics just beginning to be explored, as it involves striking a balance between task difficulty [1, 2] and instance difficulty [3, 4].
> In this study, we have gained valuable insights, demonstrating that significant performance improvements are consistently achieved across diverse domains and in the presence of various types of noise, even when the number of fragments is fixed at four, as it was throughout all our experiments detailed in the main manuscript. Furthermore, to provide a more comprehensive understanding of the impact of fragment numbers, we conducted a detailed analysis of their effects in Figures 7 and 8.
>
> [1] Mao, Y., et al. Metaweighting: learning to weight tasks in multi-task learning. ACL, 2022.
>
> [2] Guo, M., et al. Dynamic task prioritization for multitask learning. ECCV, 2018.
>
> [3] Ethayarajh, K., et al. Understanding dataset difﬁculty with V-usable information. ICML, 2022.
>
> [4] Baldock, R., et al. Deep learning through the lens of example difﬁculty. NeurIPS, 2021.

---

> ### Author Response · Authors · 2023-11-20
> **Official Comment by Submission3113 Authors**
>
> **Q4. Possible Bias during Selection (e.g., selection bias to certain class or feature, selection bias to boundary labels )**
>
> With respect to the first bias mentioned, “maybe samples whose features are severely biased to one class will be easily sampled,” FragSel’s algorithmic design already addresses them effectively. It does so by not relying solely on a single predictor or the confidence of predictions, but by considering the self and neighbor agreements of non-overlapping mixtures.
>
> Furthermore, below, we demonstrate the robust performance of FragSel in addressing the suspected bias that "if samples are located between two fragments, they may not be selected even if they are clean." We establish this by comparing the statistics of samples between two fragments, which we henceforth refer to as “boundary” samples, and the rest as “non-boundary” samples.
> For IMDB-Clean-B, we define the boundary as the ages immediately to the left/right of each fragment boundary, and for SHIFT15M-B, the boundary as the bins immediately to the left/right of each fragment boundary by binning the dataset using the same criteria during data curation. (Section D.1 Data Curation Detail). And we define the remaining ages/bins as non-boundary.
>
> We gently remind that relying solely on a "clean sample selection rate" for evaluating noisy label filtering is an insufficient metric, as previously discussed in Section 4.2 Evaluation Metrics. We reiterate that any evaluation metric should take into account the severity of noise. Therefore, we conduct an analysis of how the difference between boundary and non-boundary samples, in terms of both the selection rate and ERR, is distributed across eight experimental configurations. Specifically, the distribution of difference in the Selection rate (Table 1) yields an average mean of 2.29% with a standard deviation of 1.32%. Similarly, for ERR (Table 2), the distribution showcases an average mean of 4.12% with a standard deviation of 2.43%. This substantiates that Fragsel consistently performs robust sample selection, irrespective of the boundary. This confirmation is based on the observation that the performance difference at boundary regions does not significantly deviate from the performance exhibited across the non-boundary regions.
>
> Note that aside from the specific biases mentioned by the reviewer, our method does not inherently include balanced sampling procedures, which may lead to biased (imbalanced) sampling with respect to the overall label distribution. To mitigate this potential issue, two viable solutions can be contemplated:
>
> - **Post-Selection Balancing**: Balancing based on labels can be executed subsequent to the initial sample selection. This approach involves adjusting the selected samples to achieve a more balanced representation of labels.
>
> - **Utilization of Established Imbalanced Regression Techniques**: Alternatively, well-established imbalanced regression techniques [1, 2, 3, 4] can be applied during the subsequent training phase of the downstream regressor. Similar to the application of techniques such as cmixup and co-teach, these methods can be seamlessly integrated into our framework.
> These considerations underscore our commitment to addressing potential biases and ensuring the robustness of our approach in handling imbalanced label distributions.
>
> Table 1: Selection rate
> |data|IMDB-Clean-B||||||SHIFT15M-B||||
> |---|---|---|---|---|---|---|---|---|---|---|
> |noise|20%|40%|60%|80%|   |  |20%|40%|60%|80%|
> |boundary|79.55%|65.31%|55.98%|65.96%|   |  |31.90%|39.01%|55.44%|82.26%|
> |non-boundary|80.18%|66.86%|54.64%|61.16%|   |  |30.16%|36.63%|50.42%|82.91%|
> |difference|0.63%|1.55%|1.34%|4.80%|   |  |1.74%|2.38%|5.02%|0.65%|
>
> Table 2: ERR
> |data|IMDB-Clean-B||||||SHIFT15M-B||||
> |---|---|---|---|---|---|---|---|---|---|---|
> |noise|20%|40%|60%|80%|  |  |20%|40%|60%|80%|
> |boundary|31.90%|39.01%|55.44%|82.26%|  |  |44.52%|56.80%|60.27%|72.58%|
> |non-boundary|30.16%|36.63%|50.42%|82.91%|  |  |39.74%|52.86%|53.39%|65.04%|
> |difference|1.74%|2.38%|5.02%|0.65%|  |  |4.78%|3.94%|6.88%|7.54%|
>
> [1] Yang, Y., et al. Delving into deep imbalanced regression. In ICML, 2021.
>
> [2] Gong, Y., et al. Ranksim: Ranking similarity regularization for deep imbalanced regression. arXiv preprint arXiv:2205.15236, 2022.
>
> [3] Wang, Z., & Wang, H. Variational imbalanced regression: Fair uncertainty quantification via probabilistic smoothing. In NeurIPS, 2023.
>
> [4] Ren, J., et al. Balanced mse for imbalanced visual regression. In CVPR, 2022.

---

> ### Comment · Reviewer_PSFZ · 2023-11-21
>
> Thank the authors for their sincere efforts to relieve my concerns. Below are my answers.
>
> 1. Thank the authors for reflecting my opinion on figure 1.
> 2. I think the ERR is conceptually similar to recall. (selected true/total true data). Don't we need precision-like measure? e.g. (selected true)/(selected) I think for selecting samples, its purity would be important.
> 3. Sorry, I think I cannot find it, can the authors compare ERR and MRAE? I ask this question to see the correlation between ERR(the authors suggested) and MRAE(the original metric)
> 4. Regarding Q1 and Q2, I wanted to know the method proposed in this paper theoretically can prove the improvement.

---

> > ### Author Response · Authors · 2023-11-22
> > **Official Comment by Submission3113 Authors**
> >
> > **Q4. Regarding Q1 and Q2, I wanted to know the method proposed in this paper theoretically can prove the improvement.**
> >
> > Several theoretical aspects contribute to the improved performance of FragSel. In our previous response, we believed the reviewer was primarily interested in the theoretical underpinnings of fragmentation! In the following sections, we will reorganize and present other theoretical aspects present in the original manuscript as well as other responses.
> >
> > 1. Theoretical benefits of Mixture models resulting from Fragmentation and Neighborhood Jittering:
> >
> >     FragSel operates by partitioning data samples into fragments and leveraging trained feature extractors for sample selection through collective modeling. We conceptualize this as a Mixture-of-Experts (MoE) model, wherein individual experts specialize in specific problem subspaces through data partitioning [1, 2]. MoEs possess theoretically advantageous properties with respect to computational scalability and reduction of output variance [1], contributing to the enhancements observed in FragSel. It is noteworthy that since each network is trained on a distinct training set, MoE effectively mitigates concurrent failures, thereby preventing error propagation among networks and ultimately improving the generalization performance of FragSel as well[3].
> >
> >     Additionally, our Neighborhood Jittering leads to a Partially Overlapping Mixture Model [4], theoretically enabling the modeling of significantly richer and more intricate hidden representations by accommodating multi-cluster membership, ultimately enhancing the selection and overall performance of FragSel.
> >
> > 2. Theoretical justification of why FragSel-D (the discriminative) based model performs superior to FragSel-R (the regressive).
> >
> >     During the learning process, deep neural networks aim to maximize the mutual information between the learned representation, denoted as $Z$, and the target variable, denoted as $Y$. The mutual information between these two variables can be defined as $I(Z; Y) = H(Z) - H(Z | Y)$. A high value of $I(Z; Y)$ is indicative of a high marginal entropy $H(Z)$. Achieving this dual objective is accomplished by classification [5]. However, Zhang et al. (2023)[6] have shown that regression primarily focuses on minimizing $H(Z | Y)$ while disregarding $H(Z)$. This results in a relatively lower marginal entropy for the learned representation $Z$ and ultimately performance deficits in comparison to classification.

---

> > ### Author Response · Authors · 2023-11-22
> > **Official Comment by Submission3113 Authors**
> >
> > 3. Theoretical justification that Contrastive Fragmentation-based noisy label training does allow noisy sample identification.
> >
> >     Previously, [7] demonstrated that a binary classifier trained on noisy labels can effectively indicate the cleanliness of training data labels. Given that our methodology involves binary classification for contrasting fragment pairs, a similar property holds true with minor adjustments.
> >
> >     In Theorem 1 of [7], it is asserted that when the noisy classifier exhibits low confidence, the label is likely to be noisy with bounded probability. This is substantiated by examining the true conditional probability $\eta(x)$, Bayes optimal classifier, Tsybakov condition, transition probability, noisy classifier's prediction, and other factors. Our approach can follow the proof by simply substituting the clean and noisy label $y^\text{gt}, y$ with the clean and noisy fragment id $f^\text{gt}, f$, resulting in the assertion that the noisy binary classifier learned from contrastive pairing can assess the cleanliness of noisy labels.
> >
> >     Furthermore, even though the Tsybakov condition, which posits that the margin region near the decision boundary has a bounded volume, was assumed in [7]'s proof, the design of contrastive fragmentation can strengthen this condition. This occurs as contrastive fragmentation enforces a margin between paired fragments, creating a distinct gap in label space between them.
> >
> >     To elaborate briefly, consider a label space fragmented into four fragments (i.e., $F=4$), each covering label ranges $(y_0^L, y_0^R), (y_1^L, y_1^R), (y_2^L, y_2^R), (y_3^L, y_3^R)$. Introducing symmetric noise at a rate $\sigma$ and pairing fragments $(0, 2)$ based on noisy fragment ids $f$, the data distribution of post-contrastive fragment pairing becomes $\text{Pr}(f^\text{gt}=0)=\text{Pr}(f^\text{gt}=2)=\frac{1}{2}-\frac{1}{4}\sigma$ and $\text{Pr}(f^\text{gt}=1)=\text{Pr}(f^\text{gt}=3)=\frac{1}{4}\sigma$. (Note that samples with clean fragment ids $(1,3)$ exist because the pairing is performed based on the noisy fragment ids $f$.) Then, assuming the conditional probability of a sample $(x^{(i)}, y^{gt,(i)})$ follows the relative distance of the label to each fragment as $\eta(x^{(i)}) = \text{Pr}(f^\text{gt}=2|x^{(i)}) = \frac{\max(\min(y^{gt,(i)}, y_2^L) - y_0^R, 0)}{y_2^L - y_0^R}$, we can demonstrate that the Tsybakov condition is satisfied for data distributed in label space $y^\text{gt}\in(y_1^L, y_1^R)$. Specifically,  $\text{Pr}\left[|\eta(x) - \frac{1}{2}| \leq t\right] = \text{Pr}\left[\left|\frac{y-y_0^R}{y_2^L-y_0^R}-\frac{1}{2}\right|\leq t\right] = \frac{\sigma}{4} \times 2t = \frac{\sigma}{2} t$ when $t < 0.5$. (Detailed steps are omitted for brevity; we can provide further details upon request.) This supports the validity of the Tsybakov condition assumption in our approach.
> >
> >
> > [1] Yuksel, S. E., et al. Twenty years of mixture of experts. Transactions on neural networks and learning systems, 2012.
> >
> > [2] Masoudnia, S., & Ebrahimpour, R. Mixture of experts: a literature survey. Artificial Intelligence Review, 2014.
> >
> > [3] Sharkey, A. J., & Sharkey, N. E. Combining diverse neural nets. The Knowledge Engineering Review, 1997.
> >
> > [4] Heller. K. A., & Ghahramani. Z. A. Nonparametric bayesian approach to modeling overlapping clusters. Artificial Intelligence and Statistics, 2007.
> >
> > [5] Boudiaf, M., et al. A unifying mutual information view of metric learning: cross-entropy vs. pairwise losses. ECCV, 2020.
> >
> > [6] Zhang, S., et al. Improving deep regression with ordinal entropy, ICLR, 2023.
> >
> > [7] Zheng, S., et al. Error-Bounded Correction of Noisy Labels, ICML, 2020.

---

> ### Author Response · Authors · 2023-11-22
> **Official Comment by Submission3113 Authors**
>
> We appreciate your response and are genuinely thankful, particularly for your valuable time and efforts. We are more than happy to provide clarification or address any concerns you may have. Also, we will incorporate the pertinent updates in the forthcoming revision!
>
> **Q2&3. I think the ERR is conceptually similar to recall. (selected true/total true data). Don't we need precision-like measure? e.g. (selected true)/(selected) I think for selecting samples, its purity would be important. Analyze ERR alongside MRAE.**
>
> Yes! we can certainly identify some similarities between Error Residual Ratio (ERR) and Recall or Precision; however, there are also significant differences. Given that noisy regression labels exhibit varying degrees of noise, the evaluation metric must account for the severity of the error among the selected samples (i.e., the level of cleanliness). By contrast, in classification tasks, a simple count of clean samples among the selected would suffice. If we were to apply this logic directly to calculate Recall or Precision, the numerator selected true (True Positives) would become 0, as the sum of errors among true (clean) samples would be 0!
>
> Hence, our ERR takes into account the relation between the _average selected error_, $\frac{1}{|\mathcal{S}|}\sum_s^{|S|} |y_s - y^\text{gt}_s|$ and the _average dataset error_$\frac{1}{|\mathcal{D}|}\sum_d^{|\mathcal{D}|} |{y}^\text{ }_d - y^\text{gt}_d|$. This ratio can be interpreted as a composite measure reflecting both precision and recall. The _average selected error_ serves as an indicator of the purity or precision of the selected data, while the _average dataset error_ represents the total true error data, forming the denominator of the recall metric.
>
> Furthermore, we can compare ERR with Mean Relative Absolute Error (MRAE). However, we would like to remind that it is a common practice to consider the selection rate [1-3] together in order to simultaneously regard both quantity and quality when assessing the performance of an algorithm for optimality.
>
> Initially, we depicted the selection rate and ERR in [Figure 5](https://i.imgur.com/KajyrFY.png) (main manuscript) and [Figure 18](https://i.imgur.com/Juo1eDq.png), and [19](https://i.imgur.com/idhof9K.png) (Appendix). In order to enhance our comprehension of the relationship in conjunction with MRAE, we have integrated MRAE into these figures and conducted a comprehensive analysis below.
>
>
> As mentioned in Section 4.2, the ideal scenario for selection and refurbishment methods involves achieving a high selection rate while maintaining a low ERR, resulting in a reduced mean relative absolute error (MRAE). We examine the relationship between the selection rate, ERR, and MRAE based on Figure 5. As training progresses, FragSel and other selection methods (CNLCU-H, BMM, DY-S) approach the ideal condition, resulting in an improving trend in MRAE. FragSel, in particular, comes closest to the ideal scenario, resulting in superior MRAE performance.
>
> The most unfavorable scenario arises when there are low selection rate coupled with a high ERR, as exemplified in Figure 5(a), which is connected to a relatively worse MRAE.
>
> The scenarios of the low selection rates with low ERR and the high selection rates with high ERR can be further examined using CNLCU-H and BMM. CNLCU-H demonstrates superior selection quality in terms of ERR, while BMM exhibits a higher quantity in the selection rate. This quality/quantity trade-off is linked to the observation that CNLCU-H and BMM show similar MRAE performance in Figure 5(a). Additionally, Figure 5(b) reveals that the selection rate gap widens, while the ERR gap narrows when compared to Figure 5(a). This is associated with BMM outperforming CNLCU-H in terms of the MRAE.
>
> It's important to note that, rather than employing the selection rate and ERR as indicators for MRAE, as discussed above, these metrics offer valuable insights when assessing selected or refurbished samples directly independent of any potential regularizing effects introduced by the underlying regression model.
>
> [1] Wu, P., et al. A Topological Filter for Learning with Label Noise. NeurIPS, 2020.
>
> [2] Song, H., et al. SELFIE: Refurbishing unclean samples for robust deep learning. ICML, 2019
>
> [3] Patel, D. & Sastry, P. S. Adaptive Sample Selection for Robust Learning under Label Noise. WACV, 2023

---

### Official Review · Reviewer_3QwR · 2023-11-02

**Soundness:** 3 good
**Presentation:** 3 good
**Contribution:** 2 fair
**Rating:** 5
**Confidence:** 4

**Summary:**

Built upon the assumption that samples with similar labels tend to share relevant features, the authors propose a novel framework to model regression data collectively. They achieve this by transforming the data into disjoint yet contrasting fragmentation pairs, which utilize a mixture of neighboring fragments to identify noisy labels. This identification is carried out through an agreement among neighbors within both the prediction and representation spaces. Experimental results, demonstrated on four benchmark datasets, underscore the efficacy of the proposed framework in handling synthetic label noise.

**Strengths:**

(1) The exploration of the problem concerning noisy-labeled regression is both intriguing and practically significant.

(2) The proposed method organizes data samples into clusters and capitalizes on neighborhood information, which is well-grounded for identifying noisy labels.

(3) Tailored for noisy regression labels, the authors introduce a new metric called Error Residual Ratio for evaluating selected or refurbished samples.

(4) Empirical efforts showcased the effectiveness of the proposed method, as well as evaluated the performance of certain baseline methods (previously applied in robust classification tasks) in addressing noisy label regression.

**Weaknesses:**

(1) The presentation could be further improved to help readers capture the proposed method. [please refer to Questions Q1, Q2]

(2) When checking the empirical performances of the proposed method (FragSel-R v.s. FragSel-D), it seems that the role of classification-based feature extractor has much larger effect on the performance than the regression based method. And the performance of FragSel-R is not consistently better than baselines.

**Questions:**

(Q1) Is there a rationale behind the authors' choice to split fragments based on equal length rather than equal size? For instance, considering the age distribution of heart disease, it may be less common in children, resulting in fewer cases. Would splitting the data in intervals of 0-10, 10-20, etc., be suitable given such disparities?

(Q2) In step 2 of the proposed contrastive fragmentation algorithm, for completing the graph, could authors explain a bit more about the whole process, i.e., why the edge weight is decided by the distance between the closest samples of the two fragments, instead of the distance between two centroids.

(Q3) Regarding the performance of FragSel-R,-D, it seems that the classification-based feature extractor has much better effect than the regression based method. And the performance of FragSel-R is not consistently better than baselines.

(Q4) In Appendix Figures 7 and 8, visualizing the baseline performance, i.e., F=1 (without employing FragSel), alongside the other results could provide a more direct understanding of how the number of fragments impacts performance. This comparison could offer more insightful conclusions.

(Q5) While the authors assert that the sole hyperparameters of the framework are the number of fragments (F), the parameter K utilized for KNN-based prediction, and the extent of jittering applied for regularization, the influence of each on the results presented in Table 1 remains unclear. Could the authors provide additional insight into how these hyperparameters affect the outcomes?

---

> ### Author Response · Authors · 2023-11-20
> **Official Comment by Submission3113 Authors**
>
> **Q1&Q2. Why Fragment based on Equal Length than Equal Size? Why edge weights via distance between the closest labels of two fragments?**
>
> The foundational principle underlying FragSel is to maximize the contrast or distances between fragments, as it has been empirically demonstrated to be essential for robust training and generalizability [1, 2, 3]. Therefore, while we could conceivably consider equal fragment lengths in our approach, using equal sizes for fragment partitioning may lead to inadequate label separation among contrasting pairs, especially within densely populated label intervals. Consequently, we have chosen to employ equal fragment lengths to fully harness the advantages of contrastive pairing. Additionally, although employing the equal size criterion could assist in addressing dataset imbalances, if they exist, this concern can also be mitigated through post-selection balancing and the utilization of established imbalanced regression techniques, as detailed in reviewer PSFZ’s Q4.
>
> For the same rationale, in the assessment of edge weights, the utilization of nearest boundaries ensures a greater contrast in comparison to the centroid distances mentioned by the reviewer.
>
> [1] Shawe-Taylor, J., & Cristianini, N. Robust bounds on generalization from the margin distribution. 1998.
>
> [2] Grønlund, A., et al. Margin-based generalization lower bounds for boosted classifiers. In NeurIPS, 2019.
>
> [3] Grønlund, A., et al. Near-tight margin-based generalization bounds for support vector machines. In ICML, 2020.
>
> **Q3(W2). Performance of FragSel-D (classification) vs. FragSel-R (regression)? FragSel-R performance consistency?**
>
> Given the empirical advantages of classification training over regression, it is currently a common practice to address regression-type problems by treating them as classification tasks through the process of binning [1-3].
>
> A theoretical explanation for the greater stability of training in classification, specifically using cross-entropy as opposed to regression (such as Mean Squared Error, MSE), as posited by [4], is rooted in the concept that deep neural networks aim to maximize the mutual information between the learned representation $Z$ and the target variable $Y$. Mutual information, denoted as $I(Z; Y )$, can be defined as the difference between the marginal entropy $H(Z)$ and the conditional entropy $H(Z | Y )$. A high value of $I(Z; Y )$ indicates that the marginal entropy $H(Z)$ is high, signifying that the features in $Z$ are diverse or spread out, while the conditional entropy $H(Z | Y )$ is low, implying that the features related to common target values are close or similar. Classification achieves both of these objectives, as demonstrated by [5]. On the other hand, [6] provides evidence that regression primarily minimizes $H(Z | Y )$ but pays less attention to reducing $H(Z)$. Consequently, the learned representations $Z$ obtained from regression tend to have a lower marginal entropy.
>
> Despite the superior performance of classification-based feature extractor training, the regression-based method offers advantages in precisely handling noise. While CE treats all errors equally without emphasizing severe misses, MSE has the capability to differentiate the severity of noise. Given the crucial importance of addressing varying degrees of noise in noisy label regression algorithms, this represents a research direction that can be effectively explored via specialized regression techniques.
>
> As an auxiliary experiment, we additionally present the performance of the co-trained FragSel-R (referred to as Co-FragSel-R). Co-FragSel-R demonstrates its superiority over the majority of baseline methods, particularly when compared to other co-trained baseline methods such as CNLCU-S, CNLCU-H, and Co-Selfie.
>
> [1] Cao, Y., et al. Estimating depth from monocular images as classiﬁcation using deep fully convolutional residual networks. IEEE Transactions on Circuits and Systems for Video Technology, 2017.
>
> [2] Liu, L., et al. Counting objects by blockwise classiﬁcation. IEEE Transactions on Circuits and Systems for Video Technology, 2019.
>
> [3] Van den Oord, A., et al. Pixel recurrent neural networks, In ICML, 2016.
>
> [4] Shwartz-Ziv, R., & Tishby, N. Opening the black box of deep neural networks via information. arXiv preprint arXiv:1703.00810, 2017.
>
> [5] Boudiaf, M., et al. A unifying mutual information view of metric learning: cross-entropy vs. pairwise losses. In ECCV, 2020.
>
> [6] Zhang, S., et al. Improving deep regression with ordinal entropy, In ICLR, 2023.

---

> ### Author Response · Authors · 2023-11-20
> **Official Comment by Submission3113 Authors**
>
> **Q4. Add F=1 to Figures 7 and 8 (Fragment number analysis)**
>
> Figures 7 and 8 in the Appendix provide a visual representation of the Selection rate, ERR, and MRAE in response to variations in the fragment number, $F$. In accordance with the reviewer’s recommendation, we have included a scenario with a small fragment number in the plots to enhance the overall comprehension of the fragment number's impact. To address scenarios with a smaller fragment number, we examine cases where $F=1$ or $2$. Initially, when $F=2$, a fragment $f$ that satisfies self-agreement (Eq. 4) does not meet the criteria for neighbor-agreement (Eq. 5), as the agreement relies on comparing the scores of fragment $f$ and its contrasting pair $f^+$. Consequently, the neighborhood agreement (Eq. 6) consistently yields a value of 0. On the other hand, defining a contrasting pair is not feasible when $F=1$. As a result, the computation of the score (Eq. 3) becomes infeasible, thereby rendering the calculation of neighborhood agreement (Eq. 6) not possible. Instead, we present a plot of the vanilla baseline in Figure 7, 8 to illustrate the case when $F=1$ without utilizing FragSel.
>
> The results reveal that the MRAE of the vanilla model initially decreases during the early epochs as it learns patterns from clean samples. However, as the model begins to memorize noisy samples, the MRAE degrades. In contrast, FragSel consistently mitigates the impact of noisy samples across all plots ($F=4, 6, 8, 10$) compared to the vanilla baseline.
>
> **Q5. Do Further Hyperparameter Analysis**
>
> In Figures 7 and 8, we already investigate different fragment numbers and the influence of each on the results in Table 1 on IMDB-Clean-B as well as SHIFT15M-B dataset. As per the reviewer’s suggestion, we also include an analysis of $K$ for KNN ([Figure. 9, 10.](https://i.imgur.com/Kyxhl3A.png)) and $J$ for jittering’s influence ([Figure. 11, 12.](https://i.imgur.com/8BPt5Oi.png)).
>
> The hyperparameter $K$ determines the number of neighbors considered when assessing self/neighbor agreement from a representation perspective. With an increase in the value of $K$, the criteria for agreement become more stringent. Consequently, as $K$ increases, a greater number of confident samples are selected, resulting in a reduction in the Selection rate, ERR. However, it is worth noting that the influence of $K$ on Mean Relative Absolute Error (MRAE) performance exhibits slight fluctuations across different datasets.
>
> The hyperparameter $J$ controls the buffer range for jittering, which, in turn, determines the level of regularization applied via neighborhood jittering. Increasing the value of $J$ results in stronger regularization, effectively preventing overfitting. However, excessive regularization, as observed when $J = 0.10$, results in adverse effects during training.
>
> Specifically, in Figure 11(a) (IMDB-Clean-B Symmetric 40%), the feature extractors exhibit similar convergence patterns when $J = 0.05$ or $J = 0.10$. Consequently, comparable performance is observed in Selection Rate and MRAE. Yet, in Figure 11(b) (IMDB-Clean-B Symmetric 40%), the ERR of $J = 0.05$ is smaller than that of $J = 0.10$, leading to improved MRAE performance for $J=0.05$.
>
> Similar effects are observed in the SHIFT15M-B dataset, as depicted in Figure 12 (SHIFT15M-B).

---

### Author Response · Authors · 2023-11-20
**Official Comment by Submission3113 Authors**

We extend our heartfelt gratitude to all the reviewers for generously dedicating their valuable time and providing invaluable insights.

As part of our efforts to enhance the quality of our work, we have made several noteworthy improvements to our manuscript (in magenta), including
- More related works (with the inclusion of an additional baseline)
- Theoretical grounding of FragSel
- Additional hyperparameter analysis
- Pseudo-code representation of FragSel’s selection process
- Additional analysis of ERR, selection rate and MRAE

Additionally, we are pleased to provide you access to our implementation, which can be found at the following anonymous GitHub link: https://anonymous.4open.science/r/ICLR24_FragSel/. We have also included a user-friendly guide to assist in the execution of FragSel for your convenience. Your feedback and comments on our work are highly appreciated.

---

### Meta-Review · Area_Chair_AKLr · 2023-12-08

**Metareview:**

This paper focuses on the problem of mitigating the side effects of noisy labels in regression.

During the rebuttal, reviewers highlighted some common strengths, specifically: 1) addressing an important problem; and 2) demonstrating empirical effectiveness.

However, several concerns were also raised during the rebuttal, and reviewers acknowledged that these concerns have not been sufficiently resolved. Specifically, it remains unclear how to properly choose the best number of fragmentations (Reviewer KgTt and Reviewer PSFZ). Additionally, 2) the methods are hard to understand (Reviewer KgTt), and 3) the evaluation metrics are not convincing enough to demonstrate the effectiveness (Reviewer PSFZ).

The AC regretfully rejects the paper for now but encourages the authors to address the aforementioned concerns. I recommend that the authors revise the paper in line with the feedback provided. I believe that this paper will be much stronger after addressing these issues.

**Justification For Why Not Higher Score:**

Overall, this is an interesting paper. The concern regarding how to properly choose the number of fragmentations has been intensively discussed. Two reviewers believe that this concern has not been well solved.  Additionally, concerns related to the rigor of the evaluation and writing quality have also been raised. We believe that the paper could be significantly strengthened by addressing these concerns in the next round.

**Justification For Why Not Lower Score:**

NA

---

### Decision · Program_Chairs · 2024-01-16

Reject